# Fluorescent Sensors for the Detection of Heavy Metal Ions in Aqueous Media

**DOI:** 10.3390/s19030599

**Published:** 2019-01-31

**Authors:** Nerea De Acha, César Elosúa, Jesús M. Corres, Francisco J. Arregui

**Affiliations:** 1Department of Electric, Electronic and Communications Engineering, Public University of Navarra, E-31006 Pamplona, Spain; cesar.elosua@unavarra.es (C.E.); jmcorres@unavarra.es (J.M.C.); parregui@unavarra.es (F.J.A.); 2Institute of Smart Cities (ISC), Public University of Navarra, E-31006 Pamplona, Spain

**Keywords:** heavy metal ions, fluorescent sensors, fluorescent aptasensors, quantum dots, organic dyes

## Abstract

Due to the risks that water contamination implies for human health and environmental protection, monitoring the quality of water is a major concern of the present era. Therefore, in recent years several efforts have been dedicated to the development of fast, sensitive, and selective sensors for the detection of heavy metal ions. In particular, fluorescent sensors have gained in popularity due to their interesting features, such as high specificity, sensitivity, and reversibility. Thus, this review is devoted to the recent advances in fluorescent sensors for the monitoring of these contaminants, and special focus is placed on those devices based on fluorescent aptasensors, quantum dots, and organic dyes.

## 1. Introduction

Monitoring the presence of contaminants in water is of general interest in order to ensure the quality of surface, ground, and drinking water [1,2]. Among the several water pollutants, such as plastic or waste [3], chemical fertilizers or pesticides [4], and pathogens [5], heavy metal ions are known for their high toxicity [6]. Although some of them are essential nutrients (for instance, iron, zinc, or cobalt), they can be toxic at higher concentrations [7]. For their part, cadmium, lead, and mercury are highly poisonous even at trace levels [8,9], showing a close association to cancer or neurodegenerative diseases [10,11]. Furthermore, heavy metal ions are non-biodegradable substances [12] and they have an accumulative effect in human body [13], where they enter, typically, through the air [14], beverages [15], and the food chain [16], in which water plays a key role. There, metal ions can be found as a result of vehicle emissions [17], batteries [18], or industrial activities [19]. Thus, their detection at low concentrations is a matter of priority for environmental protection and disease prevention as well [20].

This issue requires highly sensitive and selective devices [21,22], which can be based on different technologies; for instance, electronics [23], electrochemistry [24], or optics [25]. In particular, optical sensors present numerous attractive features: the ease of integration in microfluidic platforms [26] and the capability of monitoring hazardous environments [27] are just two of them. Among the optical sensors, fluorescent ones have gained popularity in recent years since they provide high specificity as well as low detection limits, fast response time, and technical simplicity [28,29]. Their working principle consists of the emission of light by a material (fluorophore) after being excited at lower wavelengths [30]. The intensity (or lifetime) of that emission varies with the concentration of the target analyte [31]. So far, several materials, such as porphyrins [32], metal-organic frameworks [33], DNAzymes [34], fluorescent aptamers [35], quantum dots [36], or organic dyes [37] have been developed for the monitoring of heavy metal ions in water. This review is focused on the recent advances in sensors that employ the last three kinds of materials: the first section is devoted to the different techniques based on fluorescent aptamers, the second one is dedicated to the sensors fabricated with quantum dots and, finally, the third one analyzes the devices developed using organic dyes.

## 2. Heavy Metal Ion Sensors Based on Fluorescent Aptamers

Aptamers are a type of artificial oligonucleotide (ON) sequences with the ability of specifically binding to a target molecule [38]. Among their several attractive properties for the design of sensors are good thermal stability [39], the ease of synthesis and modification [40], or their simple immobilization procedure [41]. Their high affinity and specificity toward each of their target analytes [42,43] is the most remarkable property. Consequently, aptamer-based detection techniques have emerged as very selective recognition tools [44,45,46].

For instance, thymine (T) exhibits great affinity towards mercury (II) ions, forming T-Hg^2+^-T base pairs in DNA duplexes [47], and cytosine (C) forms C-Ag^+^-C mismatches when it interacts with Ag^+^ ions [48]. Thus, since the first ON-based Hg^2+^ sensor was reported by Ono et al. [49], T-rich ON sequences have been widely employed for the selective detection of Hg^2+^ in water samples [50,51,52]. Furthermore, several selective sensors for Ag^+^ have also been reported in the literature [53,54,55]. The basis of Hg^2+^ sensors based on aptamers is the conformational change of the T-rich ON sequence, which acquires a hairpin structure due to T-Hg^2+^-T mismatches, as shown in Figure 1. As Ag^+^ ions and cytosine form C-Ag^+^-C base pairs, Ag^+^ monitoring is carried out with similar sequences than Hg^2+^, but substituting the thymine groups with cytosine ones.

Another particular case is the formation of guanine (G)-quadruplexes induced by the presence of Pb^2+^ [56], as it is depicted in Figure 2. Although other metal ions, such as K^+^, Na^+^, or Ca^2+^ can slightly influence the conformation of the G-quadruplex structure [57], G-rich aptamers show good selectivity and specificity towards Pb^2+^ [58] owing to the high binding ability between Pb^2+^ and G bases [59].

Utilizing these sensing structures, different fluorescent detection strategies have been developed, all of them determined by the conformational change of the ON sequences in the presence of the target metal ion. 

The main mechanisms for the monitoring of heavy metal ions with aptamers are shown in Figure 3. For the sake of simplicity, the sensing procedures exposed in this schematic are specific for the particular case of Hg^2+^, but they can be implemented for the detection of other metal ions as long as these compounds induce a conformational change of the aptamer. These detection procedures, as well as the labelling of the fluorophores (and, if necessary, the quenchers), depend on the utilization of one or two DNA strands, as explained below.

In the case of utilizing a single T-rich ON sequence, the luminophore is usually labeled to one of its termini (5′ or 3′). Its fluorescent emission can be quenched either by the electron transfer to the T-Hg^2+^-T mismatches (Figure 3a) [60] or by a quencher linked to the other termini (3′ or 5′) of the sensitive strand (Figure 3b) [61]: as the T-Hg^2+^-T mismatches are formed, the ON sequence acquires a hairpin structure [62], decreasing the distance between the fluorophore and the quencher. This fact promotes the energy transfer between the first one (which acts as donor) and the second one (which acts as acceptor) [63].

The third procedure (Figure 3c) consists of the detection of Hg^2+^ by competitive binding: a T-rich aptamer (labeled to the fluorophore or the quencher) is usually linked to its complementary DNA (labeled to the quencher or the fluorophore). In the presence of Hg^2+^, the complementary DNA separates from the aptamer, to which the Hg^2+^ ions bind, forming the T-Hg^2+^-T mismatches that give rise the hairpin structure [64].

Typical fluorophores that are labeled to the sensitive DNA sequences are dyes, such as 6-carboxyfluorescein, (6-FAM), carboxytetramethylrhodamine (TAMRA), or Texas Red, among others, as well as fluorescent quantum dots and up-conversion nanoparticles. Chen et al. [65] developed a mercury-mediated aptamer-beacon by labeling a 6-carboxyfluorescein (FAM) to the 5′ termini of a T-rich oligonucleotide: 5′-FAM-CGC TTG TTT GTT CGC ACC CGT TCT TTC TT-3′. In the presence of Hg^2+^, this aptamer acquired a hairpin structure due to the formation of the T-Hg^2+^-T pairs. This led to the fluorescence resonance energy transfer (FRET) from the FAM to the T-Hg^2+^-T base pairs and, consequently, to the decrease of the fluorescence intensity, as displayed in Figure 4. The sensing system exhibited a limit of detection (LOD) of 4.28 nM Hg^2+^ and a linear detection range from 14.2 nM to 300 nM Hg^2+^. Furthermore, the selectivity of the sensor for Hg^2+^ over a series of metal ions (Pb^2+^, Ag^+^, Cu^2+^, Ca^2+^, Ba^2+^, Ni^2+^, K^+^, Cd^2+^, Co^2+^, Cr^3+^, Fe^3+^, Al^3+^, Mn^2+^, and Zn^2+^) was also analyzed: it was found that not one of them presented any kind of interference, even at 16–67 times higher concentrations than that of Hg^2+^.

As explained previously, the fluorescent emission of the dyes can also be attenuated by a quencher linked to the opposite termini of the sensitive aptamer. One which has been widely employed with this aim is 4-([4-(dimethylamino)phenyl]azo)benzoic acid (DABCYL) [66], owing to its broad absorption spectrum [67]: the sensor reported by Li and co-authors [68] presented a linear calibration curve between 10 nM and 200 nM, with a LOD of 10 nM Hg^2+^. Furthermore, in order to avoid the biodegradation of the aptamer, it was encapsulated in a porous phospholipid nanoshell (PPN), allowing its utilization in human urine samples.

Additionally, by labeling FAM and DABCYL to the 5′ and 3′ terminus of a thrombin-binding aptamer (TBA), which is also a T-rich sequence, a Pb^2+^ and Hg^2+^ sensor was developed: it presented a LOD of 300 pM and 5 nM for Pb^2+^ and Hg^2+^, respectively, and linear detection ranges from 0.5 nM and 30 nM for Pb^2+^ and from 10 nM to 200 nM for Hg^2+^ [61]. As the TBA is a T- and G-rich aptamer [69], in order to develop a sensor for a specific metal ion, masking agents were used: the presence of Pb^2+^, and that of Cu^2+^, Co^2+^, Ni^2+^, Cd^2+^, Cr^3+^, Al^3+^, and Fe^3+^ ions as well, was masked by adding phytic acid to the samples. The interference of Hg^2+^ was avoided by utilizing CN^−^ and a random DNA, as can be seen in Figure 5. Regarding to the reutilization of the sensor, it provided recoveries between the 95% and the 104%.

As gold nanoparticles (Au NPs) are good energy acceptors [70], they are another kind of fluorescence quenchers [71]: by covalently linking a Hg^2+^-sensitive aptamer labeled with FAM to an Au NP, it was possible to linearly monitor the Hg^2+^ concentration from 20 nM to 1000 nM [29], with a LOD of 16 nM. Furthermore, the utilization of Au NPs helped to stabilize the aptamer and decrease the LOD [72]. The quenching effect of Au NPs has also been utilized to fabricate “turn-on” fluorescent sensors: while the sensitive aptamer is linked to an Au NP (quencher), the complementary DNA sequence is labeled with a fluorophore [73], or vice versa [74]. In the absence of mercury (II) ions, the aptamer and the complementary strand are linked, so the fluorophore and the Au NP are in proximity and fluorescence transfer occurs, resulting in a negligible fluorescent emission. In the presence of Hg^2+^, due to the high specificity of the thymine groups to this metal ion, the sensitive aptamer acquires a hairpin structure, displacing the complementary strand away from the Au NP, which leads to an increase of the fluorescence [64].

Based on the competitive binding mechanism, an optical fiber sensor for monitoring Pb^2+^ was fabricated [75]. The complementary strand was deposited onto the optical fiber: in the absence of Pb^2+^, when the Cy5.5-labeled aptamer bound to the complementary DNA, its fluorescent emission was coupled to the optical fiber. Oppositely, when Pb^2+^ was present, it induced the aptamer to form G-quadruplex structures, being detached from the complementary strand, which resulted in a decrease of the coupled fluorescent intensity, as can be observed in Figure 6.

There also exist different dyes that are sensitive to the formation of double-stranded DNA (dsDNA), such as PicoGreen [76] and SYBR Green 1 [77]. Utilizing this feature, a fluorescent assay was developed for the detection of Hg^2+^ and Ag^+^ ions utilizing two complementary strands: 5′-TTCTTTCTTCCCCTTGTTTGTT-3′ and 5′-AACAAACAAGGGGAAGAAAGAA-3′ [78]. In the absence of both of these metal ions, the two strands formed a dsDNA, which led to the fluorescence emission by PicoGreen. As the Hg^2+^ or Ag^+^ concentration increased, the number of aptamer/cDNA sequences decreased, resulting in a diminution of the PicoGreen emission. The LOD was 50 nM for Hg^2+^ and 930 pM for Ag^+^, while the linear detection ranges were 50 nM to 4 µM and 930 pM to 930 nM for Hg^2+^ and Ag^+^, respectively. Furthermore, not one of the other analyzed metal ions, including Cu^2+^, Li^+^, Zn^2+^, Na^+^, Ca^2+^, Mg^2+^, K^+^, and Pb^2+^, interfered in the measurements and, finally, the sensor presented recoveries of the 80–105% for Ag^+^ and 104–114% for Hg^2+^.

As most of the aptamer-based fluorescent sensors for heavy metal ions are focused on Hg^2+^ and Pb^2+^ ions detection, the main ones are summarized in Table 1 (Hg^2+^) and Table 2 (Pb^2+^). Due to the high affinity of T-rich and G-rich ON sequences to these ions, and their high toxicity even at trace levels, less attention has been paid to other heavy metal ions. Thus, the development of sensitive and specific ON sequences for those ions is one of the main challenges for scientists.

## 3. Heavy Metal Ion Sensors Based on Fluorescent Quantum Dots

Quantum dots (QDs) are nanocrystals [93] that exhibit interesting optical properties, such as narrow and symmetric emission spectra and broad absorption band [94]: both parameters are tunable by modifying their material, shape, and size [95]. Thus, they can be used in a wide range of applications, for instance, photovoltaic devices [96], light-emitting diodes [97], or bioimaging [98].

Fluorescent QDs can be fabricated utilizing semiconductor materials [99,100], carbon [101] or carbon derivatives [102]. Sensing devices can be developed following three main strategies, which are shown in Figure 7: direct interaction between the analyte and the QDs [103], functionalization of the QDs [104,105], and integration of the QDs with other sensory materials [106].

Among the semiconductor QDs, CdTe QDs have been widely employed for the monitoring of heavy metal ions [107,108]. Furthermore, their selectivity and sensitivity can be tuned by utilizing different capping agents [109], such as thioglycolic acid (TGA) or L-cysteine: in the first case, an electron transfer process occurs between the functional groups of TGA and Hg^2+^ ions, which quenches the luminescent intensity of the CdTe QDs. Thus, employing TGA capped CdTe QDs, it was possible to detect Hg^2+^ in the nanomolar range, from 1.25 × 10^−9^ M to 1 × 10^−8^ M, with a LOD of 3.5 × 10^−10^ M Hg^2+^, as it can be observed in Figure 8a. In the case of L-cysteine capped CdTe QDs, their interaction with Hg^2+^ ions depends on the concentration of the metal ion: for concentrations of Hg^2+^ in the picomolar range, these ions interact with the carboxylate moiety of the L-cysteine on the surface of CdTe QDs by electrostatic forces [110]. As a consequence, their luminescent intensity was linearly quenched by the Hg^2+^ ions from 5 × 10^−12^ M to 2.5 × 10^−11^ M, as it is displayed in Figure 8b. Furthermore, the LOD of this sensor was 2.7 × 10^−12^ M Hg^2+^_._ At higher concentrations of Hg^2+^, there is an electron transfer between the Hg^2+^ ions and the L-cysteine capped CdTe QDs [111] which induces not only a quenching of the luminescence, but also a red shift in the luminescence peak. Other QDs that show sensitivity to Hg^2+^ are hyperbranched-graft-copolymers-capped CdS QDs [112], L-cysteine-capped ZnS QDs [113] or polyethylene glycol-capped ZnO QDs [114].

Sometimes, these QDs are functionalized with a sensitive element. That is the case of cysteamine-capped CdTe/ZnS core-shell QDs functionalized with T-rich aptamers [115] which, as explained previously, exhibit high affinity for Hg^2+^ ions. In the absence of Hg^2+^ ions, these aptamers act as an aggregator agent, resulting in a quenching of the fluorescence. As the Hg^2+^ concentration increases, T-rich aptamers acquire a hairpin structure and are detached from the QDs, which de-aggregate, giving rise to an increase of the fluorescent intensity. The LOD of this sensor was 8 × 10^−11^ M Hg^2+^, and it was capable of detecting Hg^2+^ linearly from 5 × 10^−10^ M to 1 × 10^−6^ M Hg^2+^. Taking advantage of the high affinity of thiourea for Hg^2+^, this compound was used to modify CdSe/CdS core-shell QDs for the development of a Hg^2+^ sensor [116] with a LOD of 2.79 × 10^−9^ M and a linear detection range from 5 × 10^−9^ M to 1.5 × 10^−6^ M. Although Cu^2+^ was found as an interferent ion, its presence could be masked with potassium cyanide. Apart from that, the recoveries of the fluorescent emission after removal of Hg^2+^ were between 83.8% and 95.4%.

Carbon QDs (CQDs) are a new kind of fluorescent nanomaterials [117] that exhibit several advantages over semiconductor QDs, such as good biocompatibility [118], low toxicity [119], good aqueous solubility [120], or facile synthesis [121]. 

A common approach to tune their fluorescent properties is by doping them of other elements [122]: nitrogen (N) is the most commonly employed one [123,124,125], but boron (B), sulfur (S), and phosphorous (P) are also utilized [126,127,128]. Liu et al. [129] improved the performance of carbon dots by N-doping: although the first ones exhibited a larger linear detection range (from 6 × 10^−7^ to 1.4 × 10^−5^ M, while that of N-doped CQDs was from 0.2−8 µM), their LOD was much lower, 8.7 × 10^−8^ M, opposite to that of 2.5 × 10^−7^ M of the CQDs, as shown in Figure 9. Furthermore, the N-doping also enhanced the selectivity of the CQDs, avoiding the interference of Ag^+^, Fe^3+^, Cu^2+^ and Cd^2+^ cations. The potential use of N-CQDs in real applications was tested by determining Hg^2+^ concentrations in real water samples: in the case of mineral water, the recoveries of the N-CQDs were between 96.6% and 105.5%, while in tap water they ranged from 98.5% to 105%.

While N-dopants improve the quantum yield on CQDs [130], S-dopants are good ligands for metal ions [131]. Thus, S-doped carbon dots have been widely employed for the detection of Fe^3+^ ions [132,133,134]. One example is the sensor reported in [135], which exhibited high selectivity for Fe^3+^ at pH 0. In this acid media, the LOD of the sensor was 9.6 × 10^−7^ M, while the linear detection range was from 2.5 × 10^−5^ to 5 × 10^−3^ M.

In order to enhance the properties of CQDs, they can be doped of several compounds. N-, S-, co-doped carbon dots without any functionalization were fabricated for the linear detection of Hg^2+^ [136] between 0 and 20 µM, with a LOD of 1.7 × 10^−7^ M. Additionally, the fluorescent intensity can be linearly recovered by using cyanide anions.

A particular kind of carbon QDs are based on graphene: their features are derived from graphene and carbon nanodots [137]. Hence, their sensing properties can also be modified with dopants such as nitrogen or sulfur [138,139]. In particular, N-, S-codoped graphene QDs-based paper strips have been used in real waste water for the detection of Hg^2+^ ions [140]: as it can be observed in Figure 10a, the luminescence intensity of the QDs-coated paper strips decreased as the Hg^2+^ concentration increased from 10 to 200 µM. Furthermore, concentrations of 100 µM of other metal ions (Fe^2+^, Mn^2+^, Cr^3+^, Cd^2+^, Co^2+^, and Zn^2+^) did not present any interference, as it is displayed in Figure 10b.

Most of the sensors based on fluorescent QDs have been developed with the purpose of monitoring Hg^2+^ ions. Thus, Table 3 is focused on this kind of sensors, while devices for other metal ions, are summarized in Table 4.

## 4. Heavy Metal Ion Sensors Based on Organic Dyes

Organic dyes have been widely employed for the development of fluorescence-based sensors [166,167,168] because of their attractive features: high molar extinction coefficient [169], bright signal [170], ease of modification [171], and presence of many possible reactive sites in their skeletons [172]. For the detection of heavy metal ions, these fluorophores are modified with an ion recognition unit (ionophore), which serves as host for the target metal ion (guest) [173]. The interaction between the ionophore and the target analyte induces a modification of the photophysical features of the fluorophore that is translated into a change of its fluorescent emission [174], usually from “off” to “on”. Typically employed ionophores are crown ethers [175] and aliphatic or aromatic amines [176], which act as electron donors, that is, they quench the fluorescent emission through a photo-induced electron transfer (PET) mechanism with the fluorophore [177] in the absence of the target metal ion. However, in its presence, PET does not occur, giving rise to an enhancement of the fluorescence intensity, as shown in Figure 11.

Among all of the organic dyes, rhodamine derivatives are the most utilized ones due to their structure-dependent properties [178,179,180]. Other dyes that are also widely employed for the fabrication of fluorescent sensors are fluorescein [181,182] and coumarin derivatives [183,184].

Li and co-workers [185] developed a turn-on fluorescent probe for Hg^2+^ based on a rhodamine B derivative (rhodamine B hydroxamate spirolactam) linked to a NS_2_ unit as a receptor that detected Hg^2+^ linearly from 0 to 1.6 × 10^−5^ M with a LOD of 2.36 × 10^−6^ M. The fluorescent response of the sensor towards Hg^2+^ was not interfered by any other metal ion and the probe was regenerated by using Na_2_S. Furthermore, the potential utilization of this sensor in real applications was tested by exposing the sensor to three natural water samples to which different Hg^2+^ concentrations were added, as shown in Figure 12.

Rhod-5N is another rhodamine derivative that consists of a 5N-BAPTA ionophore linked to a rhodamine fluorophore: it exhibits a fluorescence enhancement in the presence of Hg^2+^ and Cd^2+^ [186]. Ruan et al. [187] fabricated a Hg^2+^ sensor by immobilizing it in a silica sol-gel matrix onto the end of an optical fiber: its LOD was 1.25 × 10^−7^ M Hg^2+^, and it also presented a linear detection range up to 5 × 10^−7^ M Hg^2+^.

Apart from for the detection of Hg^2+^, rhodamine derivatives have been utilized for monitoring Cu^2+^, Cd^2+^, or Pb^2+^ ions [188,189]. For instance, a probe based on rhodamine 6G and p-Cresol derivatives [190] exhibited a fluorescence enhancement under the addition of Pb^2+^ ions when it was illuminated with UV light. Furthermore, the color change promoted by Pb^2+^ allowed the naked-eye detection of that ion, as shown in Figure 13.

Regarding to fluorescein, it has been shown that the modification of its sites with different functional groups gives rise to sensors of different sensitivities to Ag^+^ ions [191]: when the 4,5-positions of fluorescein were modified with N,Se-containing receptors, the LOD of the sensor was 3 × 10^−8^ M Ag^+^, whereas the introduction of a N,S-receptor decreased it up to 4 × 10^−9^ M. In both cases, the presence of Ag^+^ ions induced the opening of the spironolactone ring, which led to the increase of the fluorescence intensity, as shown in Figure 14.

Another commonly used dye is coumarin, which has been functionalized with receptors that are sensitive to different heavy metal ions: in the case of utilizing a triazole substituted 8-hydroxyquinoline (8-HQ) receptor [192], which exhibits a high affinity towards Pb^2+^, a highly sensitive (LOD = 3.36 × 10^−11^ M) and selective sensor was developed. In the absence of Pb^2+^, due the PET mechanism from the receptor to coumarin, the fluorescent emission was weak. In the presence of that metal ion, the PET process did not occur, so the blue emission of coumarin was recovered and visually detectable.

As well as in the case of aptamer- or QDs-based heavy metal ions sensors, most of the devices based on organic dyes are devoted to the detection of Hg^2+^ ions, whereas those dedicated to the monitoring of other heavy metal ions are not so numerous.

Table 5 summarizes the sensors for Hg^2+^ detection based on organic dyes, whereas those developed for the detection of other metal ions are outlined in Table 6.

## 5. Comparison between Fluorescent Sensors for Heavy Metal Ions Based on Different Materials

Although this review is focused on those sensors fabricated with fluorescent aptamers, quantum dots, and organic dyes, other materials can be utilized for the detection of heavy metal ions, such as porphyrins and metal-organic frameworks (MOFs). Thus, in this section, a brief comparison between all the materials is carried out.

As it can be observed in Table 7, the sensors developed with fluorescent aptamers and quantum dots present the lowest limits of detection, oppositely to those fabricated with MOFs. Regarding the detection ranges, the sensors based on porphyrins and MOFs are capable of detecting heavy metal ions at higher concentration ranges (from nanomolar to hundreds of micromolar concentrations) than the rest of the sensors, which monitor concentrations from the picomolar range to the micromolar one. Although reversibility and specificity are not always analyzed, the obtained results are usually positive: the sensors recover their original fluorescence intensity once the contaminants are removed from the aqueous media, and the sensors are not or slightly affected by the presence of other heavy metal ions. 

## 6. Conclusions

As it has been shown in this review, fluorescence-based sensors exhibit interesting features for the monitoring of heavy metal ions in aqueous media: the devices presented here exhibit good sensitivity and selectivity, low detection limits as well as large detection ranges. Apart from that, some of them show recovery values close to the 100%, even after being tested in real water samples. Among the heavy metal ion species, special attention has been paid to the sensors devoted to Hg^2+^, as long as it is one of the most hazardous water pollutants and it presents an accumulative effect on human body through the food chain.

Depending on the application, different sensing materials can be utilized for the monitoring of heavy metal ions: on one side, the utilization of aptamers allows the development of sensors with low detection limits, good reversibility and outstanding specificity for Hg^2+^ or Pb^2+^, due to the high affinity that T and G bases present for these contaminants, respectively. Additionally, thanks to the functionalization of quantum dots, it is possible to fabricate sensors for monitoring a wide range of heavy metal ions: although their detection limits are not as low as those of the aptasensors, they present good selectivity and reversibility. Finally, the modification of organic dyes with ion recognition units also permits the detection of several metal ions in real water samples, presenting large detection ranges, good selectivity and reversibility. Although the three kinds of materials present appropriate features for sensing applications (low detection limits, acceptable selectivity and reversibility), those devices based on aptamers exhibit the lowest limits of detection and the highest selectivity, due to the high affinity of the T-rich and G-rich ODN sequences for Hg^2+^ and Pb^2+^, respectively. However, this specificity for these analytes does not allow their utilization for the detection of other metal ions, which can be done by QDs and organic dyes.

These facts, together with all the advantages that the optic technology presents nowadays, make custom-designed fluorescent sensors attractive tools for the monitoring of heavy metal ions in real applications.

## Figures and Tables

**Figure 1 sensors-19-00599-f001:**
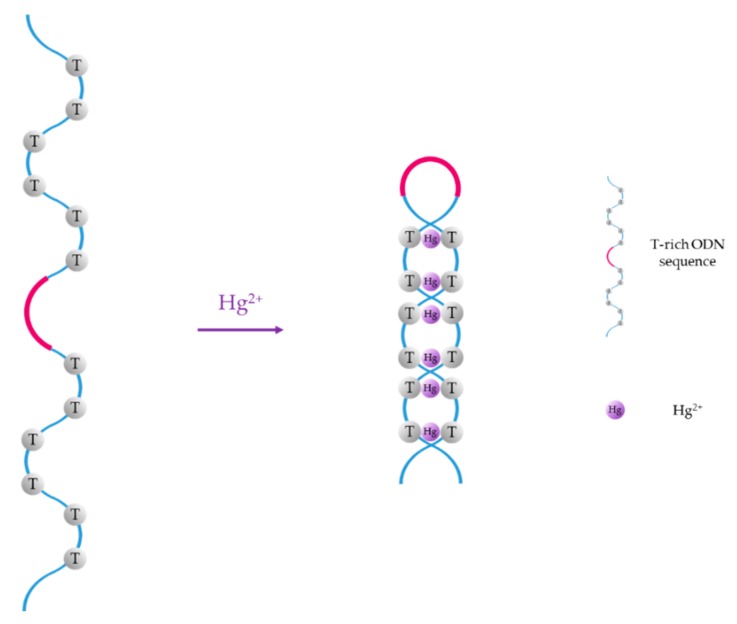
Hg^2+^-induced formation of T-Hg^2+^-T mismatches.

**Figure 2 sensors-19-00599-f002:**
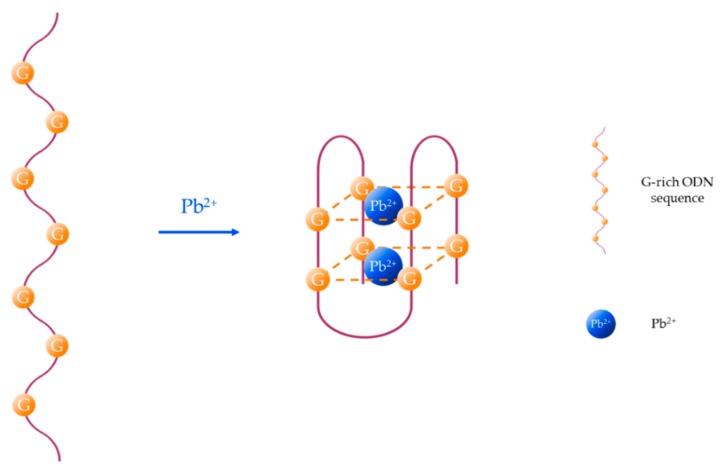
Pb^2+^-induced formation of G-quadruplex structures.

**Figure 3 sensors-19-00599-f003:**
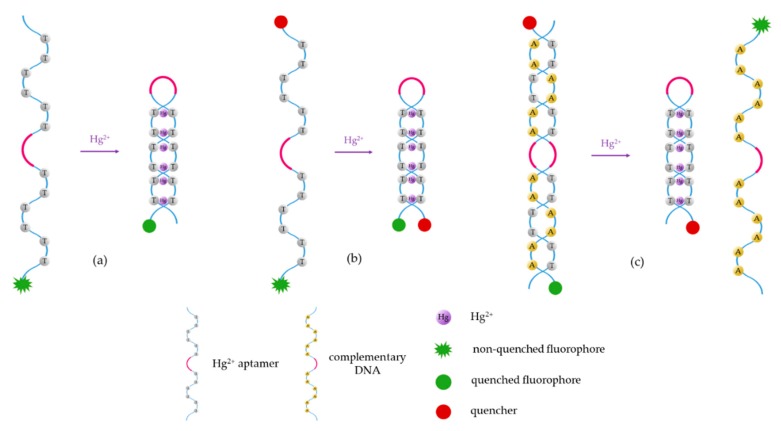
Schematic of the different Hg^2+^ sensing procedures utilizing fluorescent aptasensors: (**a**) the fluorophore is labeled to the sensitive aptamer, (**b**) the fluorophore and the quencher are linked to the two terminis of the aptamer, and (**c**) the fluorophore and the quencher are labeled to the aptamer and the complementary DNA, or vice versa. These sensing procedures can be applied to detect other heavy metal ions just by substituting the T-rich sequences by the appropriate ON sequence.

**Figure 4 sensors-19-00599-f004:**
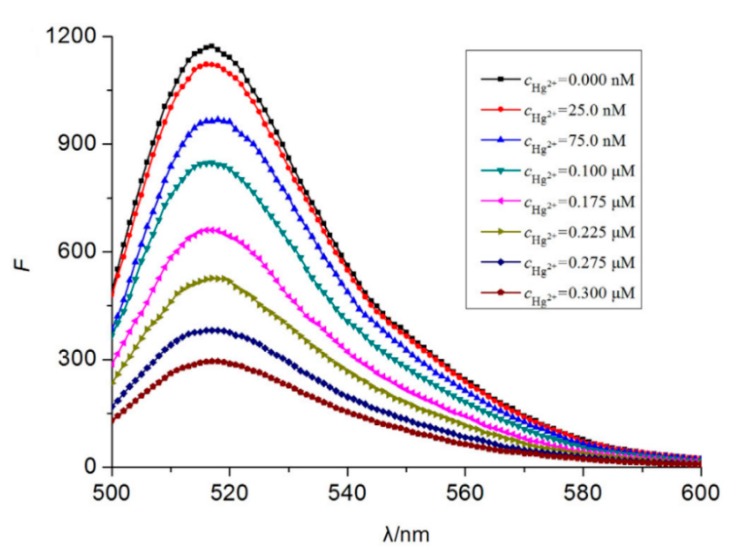
Fluorescence spectra of 5′-FAM-CGC TTG TTT GTT CGC ACC CGT TCT TTC TT-3′ for different Hg^2+^ concentrations. Reprinted with permission from [65].

**Figure 5 sensors-19-00599-f005:**
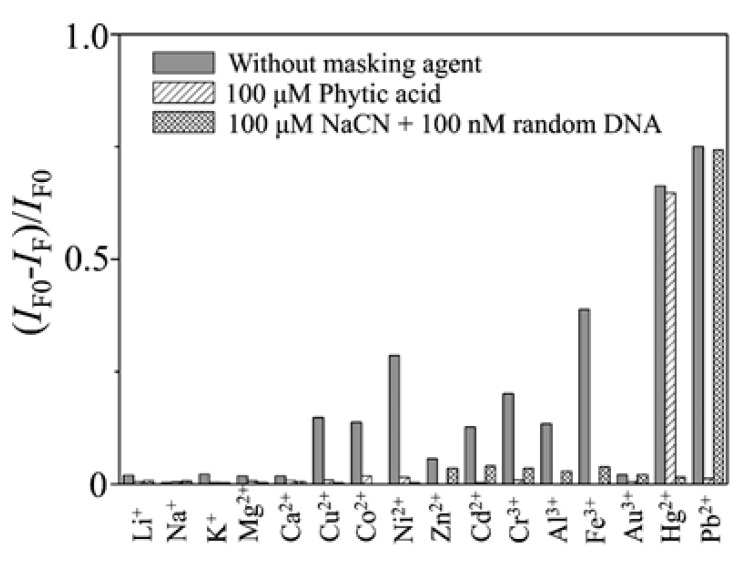
Quenching efficiency (*I*_F0_ − *I*_F_)/*I*_F0_ of the fluorescence intensity (518 nm) of the TBA-based sensor in the presence of metal ions in the absence and presence of masking agents. Concentrations: Hg^2+^, 1.0 μM; Pb^2+^, 100 nM; Li^+^ and Na^+^, 100 μM; K^+^, Mg^2+^, and Ca^2+^, 10 μM; other ions, 1.0 μM. Reprinted with permission from [61]. Copyright 2009 American Chemical Society.

**Figure 6 sensors-19-00599-f006:**
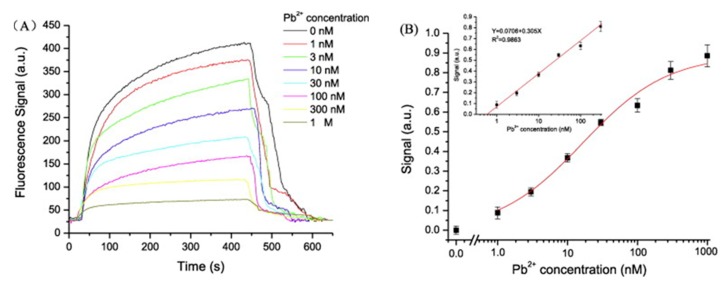
(**A**) Response curve for different Pb^2+^ concentrations in the presence of the complementary DNA. (**B**) Logarithmic calibration curve of the sensor. Reprinted with permission from [75].

**Figure 7 sensors-19-00599-f007:**
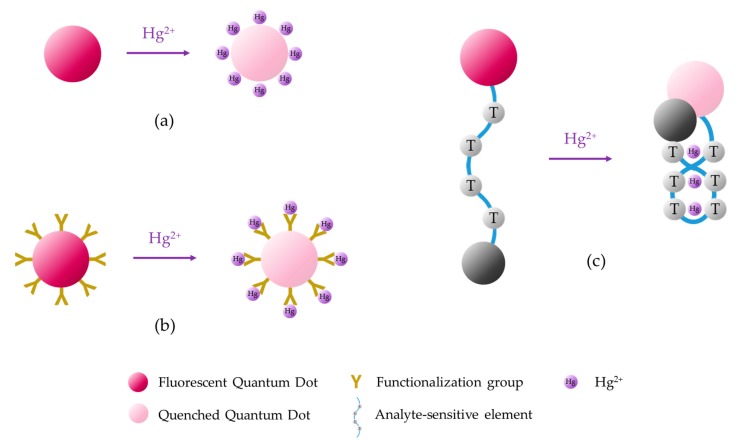
Sensing mechanisms based on fluorescent quantum dots: (**a**) direct interaction between the analyte and the QDs, (**b**) interaction of the analyte with the functionalized QDs, and (**c**) integration of the QD with another sensory material.

**Figure 8 sensors-19-00599-f008:**
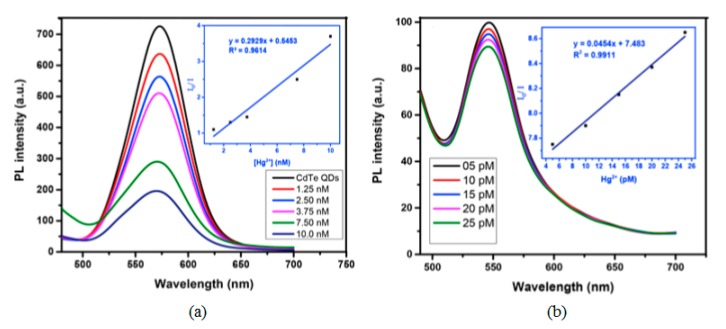
Fluorescence spectra of (**a**) TGA and (**b**) L-cysteine capped CdTe QDs for Hg^2+^ concentrations in the nanomolar and picomolar ranges, respectively. Reprinted with permission from [109].

**Figure 9 sensors-19-00599-f009:**
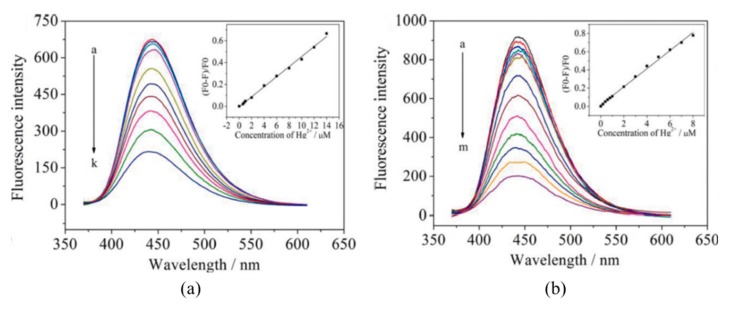
Fluorescence spectra of 10 μg/mL carbon QDs (**a**) and N-carbon QDs (**b**) upon addition of different concentrations of Hg^2+^ (a–k: 0, 0.6, 0.8, 1, 2, 4, 6, 8, 10, 12, and 14 μM; a–m: 0, 0.2, 0.4, 0.6, 0.8, 1, 2, 3, 4, 5, 6, 7, and 8 μM). The linear calibration ranges of each one of the sensors are shown in the insets. Reprinted with permission from [129].

**Figure 10 sensors-19-00599-f010:**
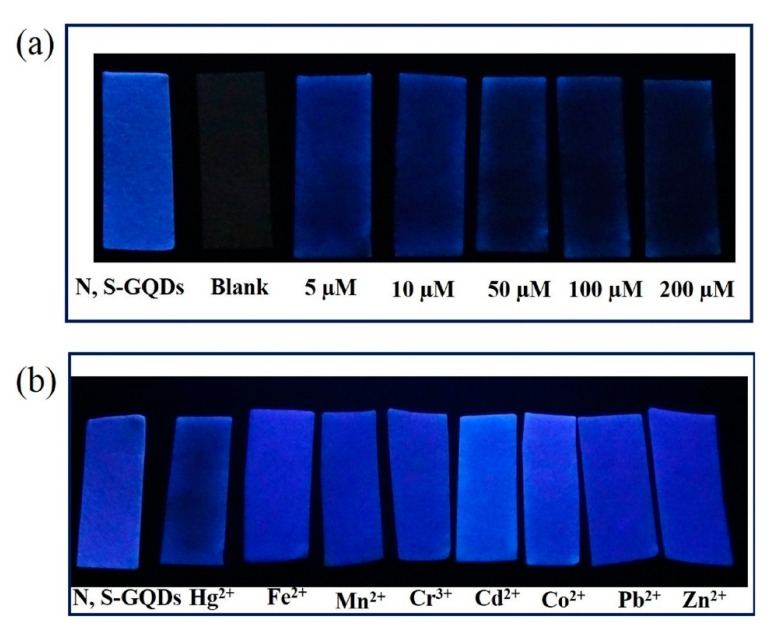
(**a**) Luminescence of blank paper strips and paper strips coated with N-, S-codoped graphene QDs exposed to different Hg^2+^ concentrations and (**b**) in the presence of different metal ions in a 100 µM concentration. Reprinted with permission from [140].

**Figure 11 sensors-19-00599-f011:**
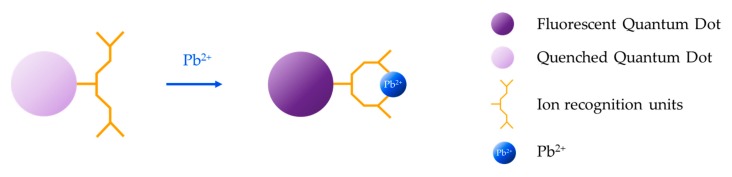
Sensing mechanism based on organic dyes modified with an ion recognition unit: the interaction between the ionophore and the target analyte induces a change in the fluorescent emission of the fluorophore.

**Figure 12 sensors-19-00599-f012:**
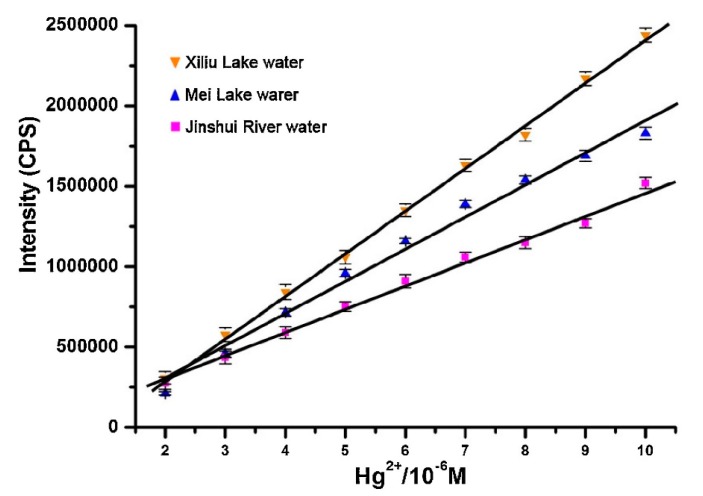
Hg^2+^ detection carried out by the rhodamine B derivative in three different natural water samples. Reprinted with permission from [185].

**Figure 13 sensors-19-00599-f013:**
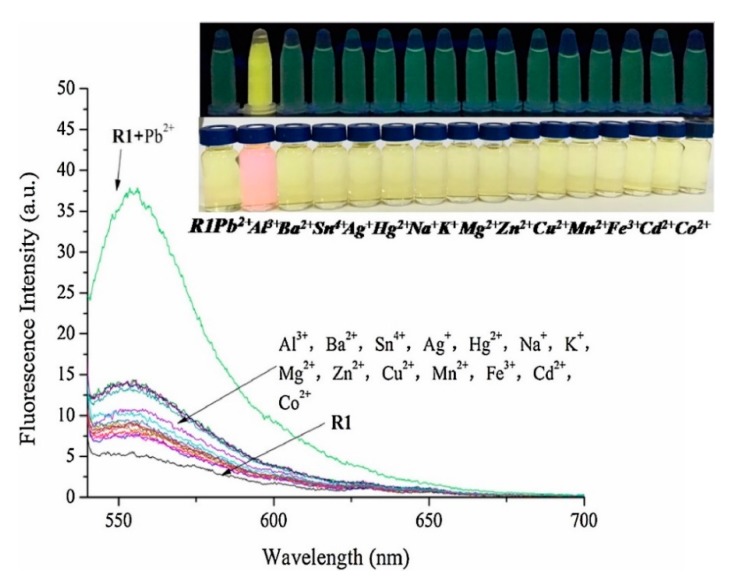
Fluorescence spectra of the probe based on rhodamine 6G and p-Cresol derivatives in the presence of Pb^2+^ (1 × 10^−5^ M) and other metal ion (5 × 10^−6^ M). The inset shows the fluorescence under UV illumination and the color change of the probe upon the addition of Pb^2+^ and other metal ions. Reprinted with permission from [190].

**Figure 14 sensors-19-00599-f014:**
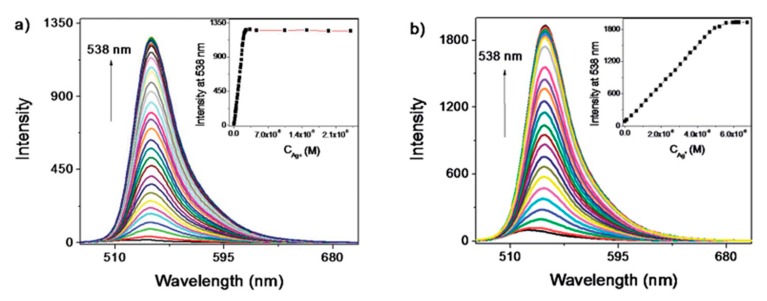
Fluorescence spectra of N,S- and N,Se-modified fluorescein sensors for different Ag^+^ ions in ethanol. Reprinted with permission from [191]. Copyright 2014 Japan Society for Analytical Chemistry.

**Table 1 sensors-19-00599-t001:** Hg^2+^ fluorescent sensors based on aptamer detection.

Analyte	ON Sequence	Fluorophore	Quencher	Detection Range	LOD	Reversibility	Interferent Analytes	Aqueous Media	Observations	Ref.
Hg^2+^	5-′FAM-CGC TTG TTT GTT CGC ACC CGT TCT TTC TT-3′	FAM		14.2 × 10^−9^ to 3 × 10^−7^ M	4.28 × 10^−9^ M	Not studied	Negligible influence	Tris–HCl buffer (10 mM, pH 8.5)		[65]
Hg^2+^	5′-NH_2_-(CH_2_)_6_- TTCTTTCTTCCCTTGTTTGTT	SYBR Green I		1 × 10^−9^ to 1 × 10^−2^ M	Not studied	93–110%	Not studied	Tris nitrate buffer (pH 8.0, 20 mM)		[79]
Hg^2+^	5′-NH_2_-(CH_2_)_6_-TTCTTTCTTCGCGTTGTTTGTT-3′	Graphene oxide (GO) sheets		1 × 10^−9^ to 50 × 10^−9^ M	9.2 × 10^−10^ M	Not studied	Negligible influence	phosphate-buffered (PBS) saline (10 mM, pH = 7.0)		[80]
Hg^2+^	5′-NH_2_-TTCTTCCCCTTGTT-3′	graphite carbon nitride (g-C_3_N_4_) sheets		5 × 10^−10^ to 1 × 10^−6^ M	1.7 × 10^−8^ M	98.3–110.8%	Cu^2+^, Fe^3+^. Ag^+^	Detection range, LOD and interferent analytes calculated in Tris-HCl (pH 7.6, 25 mM) buffer containing 150 mM NaClReversibi-lity studied in tap water		[81]
Hg^2+^	Fam-5′-GGTTGGTGTGGTTGG-3′-DABCYL)	FAM	DABCYL	1 × 10^−8^ to 2 × 10^−7^ M	5 × 10^−9^ M	95–104%	Pb^2+^	Tris–aceta-te (pH 7.4, 10 mM)		[61]
Hg^2+^	5′-FAM-GGT-TGG-TGT-GGT-TGG-DABCYL-3′	FAM	DABCYL	1 × 10^−8^ to 2 × 10^−7^ M	1 × 10^−8^ M	Not studied	Not studied	Tris-acetate buffer (pH 7.4, 10 mM)		[68]
Hg^2+^	5′-SH-3(CH_2_CH_2_O)_6_- TCATGTTTGTTTGTTGGCCCCCCTTCTTTCTTA-3′ linked to the AuNPs	Texas Red	Au NPs	1 × 10^−11^ to 1 × 10^−6^ M	5.1 × 10^−11^ M	Not studied	Negligible influence	phosphate-buffered saline (10 mM, pH 7.0) with 0.3 M NaCl	cDNA linked to the Texas Red	[64]
Hg^2+^	5′-SH (CH_2_)_6_A_10_TTCTTTCTTCCCCTTGTTTGTT-FAM-3′	FAM	Au NPs	2 × 10^−8^ to 1 × 10^−6^ M	1.6 × 10^−10^ M	Not studied	Negligible influence	Tris–HCl buffer (25 mM, pH 8.2) containing 0.3 M NaCl	Aptamer linked to the Au NPs at the 5′ termini	[29]
Hg^2+^	5′ NH_2_- C_6_-CTA CAG TTT CAC CTT TTC CCC CGT TTT GGT GTT T-3′ linked to the NaYF4:Tm^3+^, Yb^3+^ UCNPs	NaYF_4_:Tm^3+^, Yb^3+^ UCNPs	Au NPs	2 × 10^−7^ to 2 × 10^−5^ M	6 × 10^−8^ M	95.2–108.2%	Negligible influence	Detection range and LOD studied in phosphate-buffered saline (10 mM, pH 7.4) and reversibility analyzed in milk and tap water	cDNA linked to the Au NPs	[74]
Hg^2+^	5′-NH_2_-TCATCGTTCTTTCTTCCCCTTGTTTGTT-3′ linked to the UCNPs	Mn^2+^-doped NaYF_4_: Yb, Er UCNPs	Au nanoballs	5 × 10^−8^ to 5 × 10^−7^ M	1.5 × 10^−10^ M	91.4–102.3%	Negligible influence	25 g of real samples of shrimps or fish dipped in 225 mL of PBS (pH 7.4)	cDNA-functionalized Au nanoballs	[82]
Hg^2+^	5′-SH-C_6_-TACAG TTTCA CCTTT TCCCC CGTTT TGGTG TTT-3′ linked to Au NPs	Mn:CdS/ZnS QDs	Au NPs	1 × 10^−9^ to 1 × 10^−6^ M	1.8 × 10^−10^ M	Not studied	Negligible influence	Tris–HCl (pH 7.4, 10 mM) buffer with 100 mM KCl and 1 mM MgCl_2_	cDNA:5′SH-C_6_-TGAAA CTGTA-3′ linked to Mn:CdS/ZnS	[73]
Hg^2+^	5′-SH-CGTCTTGTCGA-3′ linked to QDs	Mn-doped CdS/ZnS core/shell QDs	Au NPs	1 × 10^−9^ to 1 × 10^−8^ M	4.9 × 10^−10^ M	Not studied	Negligible influence	PBS buffer (10 mM, pH 7.4)	cDNA:5′-SH-TCGTCTTGTCG-3′ linked to the Au NPs	[83]
Hg^2+^	5′-NH_2_-(CH_2_)_6_-TTCTTTCTTCGCGTTGTTTGTT-3′ labeled to the CDs	CQDs	GO	5 × 10^−9^ to 2 × 10^−7^ M	2.6 × 10^−9^ M	94.7–109.8%	Equal amount of Fe^2+^	PBS (10 mM, pH 8.0)		[84]

**Table 2 sensors-19-00599-t002:** Pb^2+^ fluorescent sensors based on aptamer detection.

Analyte	ON Sequence	Fluorophore	Quencher	Detection Range	LOD	Reversibility	Interferent Analytes	Aqueous Media	Observations	Ref.
Pb^2+^	5′-Cy5.5-(CH_2_)_6_-GGGTGGGTGGGTGGGT-3′	Cy5.5		1 × 10^−9^ to 3 × 10^−7^ M	2.2 × 10^−10^ M	80–105%	Negligible influence	PBS (10 mM, pH 7.4)	cDNA: 5′-NH_2_-(CH_2_)_6_-TTTTTTACCCACCCACCC-3′	[75]
Pb^2+^	5′-GTGGGTAGGGCGGGTTGG-3′	SYBR Green 1		1 × 10^−8^ to 1 × 10^−6^ M	Not studied	98–102.3%	Not studied	Tris–HAc buffer (10 mM, pH 7.4)		[85]
Pb^2+^	5′-GGT TGG TGT GGT TGG-3′	PicoGreen (PG)		5 × 10^−8^ to 5 × 10^−6^ M	4.8 × 10^−9^ M	Not studied	Negligible influence	Water	cDNA: 5′-CCA ACC ACA CCA ACC-3′	[86]
Pb^2+^	FAM-5′-GGTTGGTGTGGTTGG-3′-DABCYL)	FAM	DABCYL	5 × 10^−10^ to 3 × 10^−8^ M	3 × 10^−10^ M	95–104%	Hg^2+^	Tris–aceta-te (pH 7.4, 10 mM)		[61]
Pb^2+^	5′-FAM-GGTTGGTGTGGTTGG-3′	FAM	Au NPs	1.25 × 10^−8^ to 1 × 10^−7^ M	1 × 10^−8^ M	92–112%	Slightly affected by Cu^2+^, Al^2+^ and Hg^2+^	Tris–HAc buffer (5 mM, pH 7.4)		[87]
Pb^2+^	5′-SH-3(CH_2_CH_2_O)_6_-GGAAGGTGTGGAAGG-3′ linked to the Au NPs	Cy5.5	Au NPs	1 × 10^−11^ to 1 × 10^−6^ M	2.7 × 10^−13^ M	Not studied	Negligible influence	phosphate-buffered saline (10 mM, pH 7) with 0.3 M NaCl	cDNA linked to Cy5.5	[64]
Pb^2+^	5′-ATTO647N-GGGTGGGTGGGTGGGT-3′	ATTO647N	SWNTs	0 to 1 × 10^−6^ M	4.2 × 10^−10^ M	Not studied	Negligible influence	PBS buffer (10 mM, pH 7) with 0.25 M NaCl		[88]
Pb^2+^	5′-NH2-GGGTGGGTGGGTGGGT-3′ linked to NaYF_4_: Yb, Ho UCNPs	NaYF_4_: Yb, Ho UCNPs	Au NRs	1 × 10^−10^ to 1 × 10^−7^ M	5 × 10^−11^ M	96.3–110.6%	Negligible influence	25 g of real samples of shrimps or fish dipped in 225 mL of PBS (pH 7.4)	cDNA-functionalized Au NRs	[82]
Pb^2+^	5′-NH_2_-(CH_2_)_6_-GGGTGGGTGGGTGGGT-3′	Graphene QDs	GOx	6 × 10^−10^ to 4 × 10^−7^ M	6 × 10^−10^ M	Not studied	Negligible influence	PBS buffer (10.0 mM, pH 7.4)		[89]
Pb^2+^	5′-NH_2_-(CH_2_)_6_-GGGTGGGTGGGTGGGT-3′	CdSe/ZnS QDs	GO sheets	1 × 10^−10^ to 1 × 10^−8^ M	9 × 10^−11^ M	Not studied	Negligible influence	PBS buffer (10 mM, pH 7.4)		[90]
Pb^2+^	5′-GGTTGGTGTGGTTGG-3′	perylenetetracarboxylic acid diimide (PTCDI)		4.8 × 10^−10^ to 4.8 × 10^−5^ M	4.8 × 10^−10^ M	77.2–93.4%	Negligible influence	MOPS buffer (5 mM, pH 7)		[91]
Pb^2+^	5′-/3ThioMC3-D/CGATAACTCACTATrAGGAAGAGATG-3′ linked to the GQDs	Graphene QDs	Au NPs	5 × 10^−8^ to 4 × 10^−6^ M	1.67 × 10^−8^ M	Not studied	Negligible influence	PBS buffer (5 mM, pH 7.4) with 0.1 M NaCl	5′-/5AmMC6/CATCTCTTCTCCGAGCCGGTCGA-AATAGTGAGT-3′ linked to the Au NPs	[92]

**Table 3 sensors-19-00599-t003:** Hg^2+^ fluorescent sensors based on QDs.

Analyte	QDs	Detection Range	LOD	Reversibility	Interferent Analytes	Aqueous Media	Observations	Ref.
Hg^2+^	CdTe QDs	0 to 2 × 10^−6^ M	6.23 × 10^−9^ M	96.9–99.4%	Negligible influence	Ultrapure water		[141]
Hg^2+^	capped CdTe QDs	(TGA) 1.25 × 10^−9^ to 1 × 10^−8^ M (l-cysteine) 5 × 10^−12^ to 2.5 × 10^−11^ M	(TGA) 3.5 × 10^−10^ M (l-cysteine) 2.7 × 10^−12^ M	Not studied	(TGA) not evaluated (l-cysteine) Zn^2+^, Cu^2+^	Ultrapure water	QDs capped with thioglycolic acid (TGA) or l-cysteine	[109]
Hg^2+^	Cysteamine (CA)-capped CdTe QDs	6 × 10^−9^ to 4.5 × 10^−7^ M	4 × 10^−9^ M	97–106.4%	10-fold Pb^2+^, Cu^2+^ and Ag^+^ < 7%	acetic-acetate buffer (pH 5.0)		[107]
Hg^2+^	l-Tryptophan-capped carbon quantum dots	1.1 × 10^−8^ to 4 × 10^−6^ M	1.1 × 10^−8^ M	Not studied	Negligible influence	sodium phosphate buffer (10 mM, pH 6.0)		[142]
Hg^2+^	HPEI-*g*-HPGs-capped CdS QDs	1 × 10^−8^ to 1 × 10^−4^ M	1.5 × 10^−8^ M	Not studied	Cu^2+^	Tris–HCl buffer (pH 7.4, 10 mM)		[112]
Hg^2+^	MPA coated Mn doped ZnSe/ZnS colloidal NPs	0 to 2 × 10^−8^ M	1 × 10^−10^ M	Not studied	Negligible influence	PBS (10 mM, pH 7.4)		[143]
Hg^2+^	PDDA-functionalized CdTe QDs	6 × 10^−9^ to 1 × 10^−6^ M	5 × 10^−9^ M	97.5–103%	Negligible influence	Double distilled water	PDDA eliminates the interference from Cu^2+^ and Ag^+^	[144]
Hg^2+^	TU-functionalized TGA-capped CdSe/CdS QDs	5 × 10^−9^ to 1.5 × 10^−6^ M	2.79 × 10^−9^ M	83.8–95.4%	Not studied	PBS (pH 7.73)		[116]
Hg^2+^	CdTe@SiO_2_@GQDs	1 × 10^−8^ to 2.2 × 10^−5^ M	3.3 × 10^−9^ M	107.3–108.7%	Fe^2+^, Fe^3+^	PBS (10 mM, pH 7.73)		[145]
Hg^2+^	Carbon QDs blended with Rhodamine B	1 × 10^−7^ to 4 × 10^−5^ M	3 × 10^−8^ M	94.5–957%	glutathione (GSH)	High purity water		[146]
Hg^2+^	N-doped carbon QDs	1 × 10^−7^ to 1 × 10^−4^ M	2.3 × 10^−8^ M	97.2–103.8%	GSH	Ultrapure water		[147]
Hg^2+^	N-doped carbon QDs	0.2 × 10^−6^ to 8 × 10^−6^ M	8.7 × 10^−8^ M	96.6–105.5%	Negligible influence	PBS (50 mM, pH 7)	Doping with N improves the selectivity	[129]
Hg^2+^	N-doped carbon QDs	0 to 2.5 × 10^−5^ M	2.3 × 10^−7^ M	No	Negligible influence	Ultra-pure water		[148]
Hg^2+^	N-dopped carbon QDs	1 × 10^−8^ to 1 × 10^−7^ M	2.1 × 10^−9^ M	No	Not studied	PBS (10 mM, pH 7)		[149]
Hg^2+^	N-, S-, Co- doped carbon QDs	0 to 2 × 10^−5^ M	1.8 × 10^−7^ M	No	Cu^2+^, Ni^2+^	Deionized water and filtered river water		[136]
Hg^2+^	S- and O- doped carbon nitride QDs	1 × 10^−8^ to 1 × 10^−6^ M	1 × 10^−11^ M	Not studied	Negligible influence	Double distilled water and tap water		[150]
Hg^2+^	Graphene QDs	8 × 10^−7^ to 9 × 10^−6^ M	1 × 10^−7^ M	Not studied	Ca^2+^, Zn^2+^, Fe^2+^, and Co^2+^ < 10%	Tris–HCl buffer (pH 8, 50 mM)		[151]
Hg^2+^	O-rich N-doped graphene QDs	4 × 10^−8^ to 6 × 10^−6^ M	8.6 × 10^−9^ M	86.7–103.5%	Pb^2+^, Cd^2+^, Cu^2+^, and Ni^2+^	Tris–HCl buffer (pH 8, 10 mM)		[138]
Hg^2+^	N-, S-doped graphene QDs	5 × 10^−8^ to 1.5 × 10^−5^ M	1.4 × 10^−8^ M	(96 ± 4.7)–(116 ± 3.8)%	Negligible influence	PBS buffer (100 mM, pH 7)		[140]

**Table 4 sensors-19-00599-t004:** Fluorescent sensors for different metal ions based on QDs.

Analyte	QDs	Detection Range	LOD	Reversibility	Interferent Analytes	Aqueous Media	Observations	Ref.
Pb^2+^	xylenol orange functionalized CdSe/CdS QDs	5 × 10^−8^ to 6 × 10^−6^ M	2 × 10^−8^ M	94.8–103.7%	Negligible influence	PBS (pH 6.47)		[152]
Pb^2+^	green Au NCs covalently linked to the surface of silica NPs embedded with red QDs	2.5 × 10^−8^ to 2.5 × 10^−7^ M	3.5 × 10^−9^ M	95.2–112.4%	Negligible influence	PBS (50 mM, pH 6)		[153]
Pb^2+^	S-doped graphene QDs	1 × 10^−7^ to 1.4 × 10^−4^ M	3 × 10^−8^ M	Not studied	Negligible influence	PBS (3 mM, pH 7)		[139]
Pb^2+^	Silica-coated ZnS QDs (ZnS@SiO_2_ QDs)	1 × 10^−9^ to 2.6 × 10^−4^ M	-	No	Cd^2+^	Deionized water		[154]
Pb^2+^	Flavonoid moiety-incorporated carbon QDs	1 × 10^−10^ to 2 × 10^−8^ M	5.5 × 10^−11^ M	Not studied	Negligible influence	Deionized water		[155]
Pb^2+^	CdTe QDs	2 × 10^−8^ to 3.6 × 10^−6^ M	8 × 10^−8^ M	Not studied	Negligible influence	Human serum		[156]
Cu^2+^	*N*-acetyl-l-cysteine capped CdHgSe QDs	1 × 10^−9^ to 4 × 10^−7^ M	2 × 10^−10^ M	98.3–101.6%	Ag^+^, Co^2+^, Hg^2+^	PBS (pH 9)		[157]
Cu^2+^	L-cysteine capped Mn^2+^-doped ZnS QDs	7.87 × 10^−6^ to 3.15 × 10^−4^ M	3.15 × 10^−6^ M	Not studied	Hg^2+^	Phosphate buffer (pH 7)		[113]
Cu^2+^	ligand-capped CdTe QDs (CdTe-L QDs)	(5.16 ± 0.07) × 10^−8^ to (1.50 ± 0.03) × 10^−5^ M	(1.55 ± 0.05) × 10^−8^ M	Not studied	Negligible influence	Tris–HCl buffer (pH 6.5, 10 mM)		[158]
Cu^2+^	inorganic CsPbBr_3_ perovskite QDs	0 to 1 × 10^−7^ M	1 × 10^−10^ M	Not studied	Negligible influence	Hexane		[159]
Cu^2+^	Polyethylene glycol capped ZnO QDs (PEG@ZnO QDs)	4 × 10^−9^ to 1 × 10^−5^ M	3.33 × 10^−9^ M	99.6–104.0%	Negligible influence	Detection range, LOD and interferent analytes in Ultra-pure studied water, reversibi-lity in tap water		[114]
Cu^2+^	Water-soluble silica-coated ZnS:Mn NPs (ZnS:Mn/SiO_2_)	8.16 × 10^−8^ to 4.16 × 10^−4^ M	-	94.76–105.82%	Negligible influence	Seawater		[160]
Fe^3+^	Carbon QDs	0 to 3 × 10^−4^ M	13.68 × 10^−6^ M	With ascorbic acid	Negligible influence	Ultra-pure water		[161]
Fe^3+^	CdTe QDs: (1) thioglyco-lic acid capped quantum dots (Green) (2) *N*-Acetyl-l-cysteine capped QDs (red)	0 to 3.5 × 10^−6^ M	1.4 × 10^−8^ M	Not studied	Negligible influence	Deioni-zed water		[108]
Fe^3+^	S-doped carbon QDs	2.5 × 10^−5^ to 5 × 10^−3^ M	9.6 × 10^−7^ M	Not studied	Negligible influence	Ultra-pure water	It works in strongly acid (pH < 2) solutions	[135]
Fe^3+^	N-, B-, S- doped carbon dots	3 × 10^−7^ to 5.46 × 10^−4^ M	9 × 10^−8^ M	97.98–108.55%	Negligible influence	Tris–HCl buffer (pH 7)		[162]
Hg^2+^, Pb^2+^	L-cysteine-capped CdS QDs	1 × 10^−9^ to 4 × 10^−9^ M (Hg^2+^) 3 × 10^−9^ to 1.5 × 10^−8^ M (Pb^2+^)	1 × 10^−9^ M (Hg^2+^) 1 × 10^−7^ M (Pb^2+^)	Not studied	Negligible influence	phosphate buffer (pH 7.4)		[163]
Cr(III)	ligand-coated CdTe QDs (CdTe-L QDs)	(6.78 ± 0.05) × 10^−9^ to (3.70 ± 0.02) × 10^−6^ M	(20.30 ± 0.03) × 10^−9^ M	98.32–100.50%	Negligible influence	PBS (10 mM, pH 7)		[164]
Cd^2+^	Green emitting CdSe QDs covalently linked onto red emitting CdTe QDs	1 × 10^−7^ to 9 × 10^−6^ M	2.5 × 10^−9^ M	86.5–102.6%	Negligible influence	Detection range, LOD and interferent analytes studied in Tris-EDTA. Reversibility studied in lake water and tap water		[165]

**Table 5 sensors-19-00599-t005:** Hg^2+^ sensors for different metal ions based on organic dyes.

Analyte	Organic Dye	Detection Range	LOD	Reversibility	Interferent Analytes	Aqueous Media	Observations	Ref.
Hg^2+^	Rhodamine B	1 to 5 × 10^−8^ M	-	Not studied	Negligible influence	Acetonitrile	Functionalized with 5-aminoisophthalic acid diethyl ester	[193]
Hg^2+^	Rhodamine B	0 to 7 × 10^−5^ M	-	Not studied	Zn^2+^, Fe^2+^,and Cu^2+^	Water	Functionalized with glucose	[194]
Hg^2+^	non-sulfur rhodamine derivative	0 to 5 × 10^−6^ M	2 × 10^−7^ M	Yes	Negligible influence	Acetonitrile	Functionalized with ethylene moiety	[195]
Hg^2+^	Rhodamine B (RBAI) Rhodamine 6G (RGAI)	RBAI—5 × 10^−6^ to 2.2 × 10^−5^ M RGAI—7.94 × 10^−6^ to 2.5 × 10^−5^ M	RBAI—4.23 × 10^−6^ M RGAI—6.34 × 10^−6^ M	> 90%	Negligible influence	Detection range, LOD, reversibility and interferent analytes studied in ethanol-water (4/6 *v*/*v*, 20 mM, HEPES pH 7.4). Detection also tested in living cells and mice	Functionalized with di-Aminobenzene-phenyl Isothiocyanate	[196]
Hg^2+^	Rhodamine B derivative	0 to 1.6 × 10^−5^ M	2.36 × 10^−6^ M	Yes	Negligible influence	Detection range, LOD, reversibility and interferent analytes studied in deionized water. Potential application analyzed in three natural water samples.	Functionalized with NS_2_-containing receptor	[185]
Hg^2+^	Rhodamine derivative	0 to 6 × 10^−4^ M	6.79 × 10^−6^ M	Not studied	Negligible influence	DMSO–HEPES buffer (0.02 mol/L, pH 7.4; *v*/*v* = 6:4)	Functionalized with hydroxyquinoline group	[197]
Hg^2+^	Rhod-5N	0 to 3 × 10^−7^ M	1.5 × 10^−9^ M	Not studied	Not studied	Milli-Q water	Functionalized with BAPTA	[187]
Hg^2+^	Rhodamine C	4 × 10^−7^ to 5 × 10^−6^ M	7.4 × 10^−8^ M	Yes (Na_2_S addition)	Negligible influence	buffered HEPES (20 mM, pH 7.0) water-ethanol (7/3, *v*/*v*)	synthesized by the reaction of rhodamine ethylenediamine and cinnamoyl chloride	[198]
Hg^2+^	Rhodamine B derivatives RW-1, RW-2	RW-1: 5 × 10^−7^ to 3 × 10^−6^ M RW-2: 5 × 10^−7^ to 4 × 10^−6^ M	RW-1: 2.5 × 10^−8^ M RW-2: 4.2 × 10^−8^ M	Yes	Negligible influence	4:6 CH_3_OH/HEPES buffer (*v*/*v*, 10 mM, pH 7.0)	Functionalized with a spirocyclic moiety	[199]
Hg^2+^	RR1-rhodamine–rhodanine-based	0 to 12 × 10^−6^ M	5 × 10^−9^ M	No	Negligible influence	water–ACN (60/40 *v*/*v*) mixture		[200]
Hg^2+^, Pb^2+^, Cd^2+^	rhodamine 6G hydrazide	Hg^2+^: 1 × 10^−5^ to 5 × 10^−5^ M Pb^2+^: 1 × 10^−5^ to 7 × 10^−5^ M Cd^2+^: 1 × 10^−5^ to 9 × 10^−5^ M	Hg^2+^: 1.6 × 10^−8^ M Pb^2+^: 1.2 × 10^−8^ M Cd^2+^: 4.7 × 10^−8^ M	Yes: Hg^2+^ and Cd^2+^ (with EDTA) No: Pb^2+^	Cu^2+^ and Ni^2+^ in the case of Cd^2+^ detection	HEPES buffer solution (EtOH:H_2_O = 9/1, 10 mM HEPES buffer, pH 7.2)	Functionalized with N-methylisatin	[189]
Hg^2+^	Fluorescein and rhodamine B	2.5 × 10^−7^ to 2.52 × 10^−6^ M	2.02 × 10^−8^ M	Yes	Negligible influence	Dichlorome-thane		[201]
Hg^2+^	Coumarine derivative	0 to 1.4 × 10^−5^ M	-	Yes (after TPEN incubation)	Negligible influence	Deionized water	Modified with azathia crown ether moiety	[202]
Hg^2+^	rhodol-coumarin	0 to 2.5 × 10^−5^ M	5.5 × 10^−9^ M	Not studied	Negligible influence	MeOH-H_2_O (*v*/*v* = 1:1) solution	Modified with hydrazide moiety	[203]
Hg^2+^	coumarin	0 to 4 × 10^−6^ M	1 × 10^−5^ M	No	Co^2+^, Ni^2+^ and Cu^2+^ (can me masked by using EDTA)	HEPES buffer solution (20 mM HEPES, pH 7.2, EtOH:H_2_O = 1:1, *v*/*v*)	thiosemicarbazide derivative reacts with Hg^2+^	[204]
Hg^2+^	dibenzo-18-crown-6-ether (DB18C6)	1.25 × 10^−6^ to 1.2 × 10^−4^ M	1.25 × 10^−8^ M	Not studied	Cu^2+^, Pb^2+^	Titrisol buffer (pH 7)		[205]
Hg^2+^	2-((2-(vinyloxy)-naphthalen-1-yl)methylene) malononitrile	0 to 5 × 10^−6^ M	4.31 × 10^−8^ M	Not studied	Negligible influence	PBS buffer (10 mM, pH 7.4, containing 1% CH_3_CN)		[206]
Hg^2+^	Dansyl-Met-NH_2_	1 × 10^−8^ to 6 × 10^−6^ M	5 × 10^−9^ M	Yes	Potentital interference from Fe^2+^, Pb^2+^, Cd^2+^, Pd^2+^	HEPES buffer (10 mM, pH 7.4). Potential application also studied in synthetic marine water		[207]

**Table 6 sensors-19-00599-t006:** Fluorescent sensors for different metal ions based on organic dyes.

Analyte	Organic Dye	Detection Range	LOD	Reversibility	Interferent Analytes	Aqueous Media	Observations	Ref.
Pb^2+^	Rhodamine 6G derivative	1 × 10^−8^ to 1 × 10^−5^ M	2.7 × 10^−9^ M	Yes	Negligible influence	HEPES buffer (10 mM, pH 7.4). Also tested in sea shells food.	Recognition moiety attached to the R-6G derivative	[190]
Pb^2+^	rhodamine tri methoxy benzaldehyde conjugate derivative	0 to 1 × 10^−5^ M	1.5 × 10^−8^ M	Not studied	Negligible influence	HEPES buffer solution (pH 7.54)		[208]
Pb^2+^	rhodamine hydroxamate derivative	0 to 1 × 10^−5^ M	2.5 × 10^−7^ M	Yes (adding EDTA)	Negligible influence	HEPES buffer (10 mM, pH 6.5)	Functionalized with an acyclic diethyl iminodiacetate receptor	[188]
Pb^2+^	Coumarin	0 to 2 × 10^−5^ M	1.9 × 10^−9^ M	Not studied	Negligible influence	phosphate-buffer (20 mM, 1:9 DMSO/H_2_O (*v*/*v*), pH 8.0)	Coumarin-trizaole-based receptor: (4-((1-(2-oxo-2H-chromen-4-yl)-1H-1,2,3- triazol-5-yl)methoxy)-2H-chromen-2-one)	[209]
Pb^2+^	Coumarin	6 × 10^−6^ to 2 × 10^−5^ M	3.36 × 10^−11^ M	Not studied	Negligible influence	HEPES buffer solution (CH_3_CN:H_2_O = 95:5, *v*/*v*, 10 mM, pH 7.2)	Functionalized with a triazole substituted 8-hydroxyquinoline (8-HQ) receptor	[192]
Pb^2+^	BODIPY fluorophore	5 × 10^−8^ to 2.5 × 10^−6^ M	1.34 × 10^−8^ M	Not studied	Negligible influence	PBS buffer (0.1 M, pH 7.2)	Functionalized with a polyamide receptor	[210]
Pb^2+^	1,3,6-trihydroxy xanthone	1 × 10^−5^ to 2 × 10^−4^ M	1.8 × 10^−7^ M	Not studied	-	DMSO–H_2_O solution (2:1 ratio, *v*/*v*)		[211]
Pb^2+^	2-amino-4-phenyl-4H-benzo[h]chromene-3-carbonitrile	0 to 2 × 10^−3^ M	4.14 × 10^−4^ M	Yes	Cd^2+^, Fe^3+^, Hg^2+^, Cu^2+^	Methanol		[212]
Cu^2+^	rhodamine B semicarbazide	2 × 10^−8^ to 3 × 10^−7^ M	1.6 × 10^−7^ M	Not studied	Negligible influence	Methanol–water (1:1, *v*/*v*) at pH 7		[213]
Cu^2+^	rhodamine hydroxamate derivative	0 to 1.2 × 10^−5^ M	5.8 × 10^−7^ M	Yes (Na_2_S addition)	Negligible influence	HEPES buffer (10 mM, pH 6.5) containing 1% CH_3_CN (*v*/*v*)	Functionalized with an acyclic diethyl iminodiacetate receptor	[188]
Cu^2+^	6,7-dihydroxy-3-(3-chlorophenyl) coumarin	0 to 2.5 × 10^−6^ M	3.3 × 10^−10^ M	Yes (with S^2−^)	Negligible influence	CH_3_CN/H_2_O (90:10, *v*/*v*)		[214]
Cu^2+^	Fluorescein	1 × 10^−6^ to 6 × 10^−5^ M	3 × 10^−7^ M	Not studied	Negligible influence	DMSO/HEPES solution(3:1, *v*/*v*, 1 mM, pH 7.2)	Functionalized with a pyrrole moiety	[215]
Pb^2+^, Cu^2+^	styrylcyanine dye containing pyridine	Pb^2+^: 3 × 10^−5^ to 6 × 10^−4^ M Cu^2+^: 3 × 10^−6^ to 9 × 10^−7^ M	Pb^2+^: 3.41 × 10^−6^ M Cu^2+^: 1.24 × 10^−6^ M	Not studied	Negligible influence	CH_3_CN–water mixture (9:1, *v*/*v*)		[216]
Zn^2+^	Fluorescein-coumarin conjugate	0 to 1 × 10^−5^ M	1 × 10^−7^ M	Yes	Negligible influence	HEPES buffer (water/ethanol, 1:9, *v*/*v*; 10 mM HEPES; pH 7.4)		[217]
Cd^2+^	coumarin	0 to 1.6 × 10^−5^ M	-	Not studied	Hg^2+^	Deionized water	Functionalized with a dipicolylamine receptor	[177]
Ag^+^	Fluorescein	L1: 0 to 1.98 × 10^−6^ M L2: 0 to 4.95 × 10^−6^ M	L1: 4 × 10^−9^ M L2: 3 × 10^−8^ M	Yes (Na_2_S)	Negligible influence	Ethanol	L1: functionalized with N,S- receptor L2: functionalized with N,Se- receptor	[191]
Pd^2+^	Coumarin 460	0 to 1 × 10^−5^ M	2.5 × 10^−7^ M	Not studied	Negligible influence	PBS buffer containing 1% DMSO		[218]

**Table 7 sensors-19-00599-t007:** Fluorescent sensors for heavy metal ions based on different kind of materials.

Type of Material	Sensitive Material	Analyte	Detection Range	LOD	Reversibility	Interferent Analytes	Ref.
Fluorophore-labelled aptamer	Mn^2+^-doped NaYF_4_: Yb, Er UCNPs labelled to 5′-NH_2_-TCATCGTTCTTTCTTCCCCTTGTTTGTT-3′	Hg^2+^	5 × 10^−8^ to 5 × 10^−7^ M	1.5 × 10^−10^ M	91.4–102.3%	Negligible influence	[82]
	Texas Red labelled to 5′-SH-3(CH_2_CH_2_O)_6_- TCATGTTTGTTTGTTGGCCCCCCTTCTTTCTTA-3′ linked to the AuNPs	Hg^2+^	1 × 10^−11^ to 1 × 10^−6^ M	5.1 × 10^−11^ M	Not studied	Negligible influence	[64]
	5′- Cy5.5-SH-3(CH_2_CH_2_O)_6_-GGAAGGTGTGGAAGG-3′ linked to the Au NPs	Pb^2+^	1 × 10^−11^ to 1 × 10^−6^ M	2.7 × 10^−13^ M	Not studied	Negligible influence	[64]
Quantum dots	S- and O- doped carbon nitride QDs	Hg^2+^	1 × 10^−8^ to 1 × 10^−6^ M	1 × 10^−11^ M	Not studied	Negligible influence	[150]
	Flavonoid moiety-incorporated carbon QDs	Pb^2+^	1 × 10^−10^ to 2 × 10^−8^ M	5.5 × 10^−11^ M	Not studied	Negligible influence	[157]
	Polyethylene glycol capped ZnO QDs (PEG@ZnO QDs)	Cu^2+^	4 × 10^−9^ to 1 × 10^−5^ M	3.33 × 10^−9^ M	99.6–104%	Negligible influence	[114]
Organic dyes	rhodol-coumarin	Hg^2+^	0 to 2.5 × 10^−5^ M	5.5 × 10^−9^ M	Not studied	Negligible influence	[203]
	6,7-dihydroxy-3-(3-chlorophenyl) coumarin	Cu^2+^	0 to 2.5 × 10^−6^ M	3.3 × 10^−10^ M	Yes (with S^2−^)	Negligible influence	[214]
	Rhodamine 6G derivative	Pb^2+^	1 × 10^−8^ to 1 × 10^−5^ M	2.7 × 10^−9^ M	Yes	Negligible influence	[190]
Porphyrins	5,10,15,20-tetrakis (4-sulfonatophenyl)porphyrin(TPPS)	Hg^2+^	3.3 × 10^−8^ to 3.3 × 10^−5^ M	3.3 × 10^−8^ M	Not studied	Negligible influence	[32]
	5,10,15,20-tetrakis (*N*-methyl-4-pyridyl) porphyrin (TMPyP)	Hg^2+^	5 × 10^−9^ to 1 × 10^−7^ M	1.3 × 10^−9^ M	96–105%	Slightly affected by Pb^2+^	[219]
	5,10-bis(4-aminophenyl)-15,20-diphenyl-porphyrin (BATP)	Cd^2+^	5 × 10^−8^ to 4 × 10^−6^ M	3.2 × 10^−8^ M	Yes	Slightly affected by Cu^2+^ and Hg^2+^	[220]
Metal-organic frameworks	UiO-66-PSM	Hg^2+^	0 to 7.81 × 10^−5^ M	5.88 × 10^−6^ M	96.9–100.6%	Negligible influence	[221]
	MIL-53(Al)	Fe^3+^	3 × 10^−6^ to 2 × 10^−4^ M	9 × 10^−5^ M	98–106%	Negligible influence	[222]
	UiO-66-NH_2_	Cd^2+^	1 × 10^−5^ to 5 × 10^−4^ M	3.36 × 10^−7^ M	Not studied	Not studied	[223]

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
