# Peer review of "Fluorescent Sensors for the Detection of Heavy Metal Ions in Aqueous Media"

_sensors, 2019, doi:10.3390/s19030599_

Round 1
Reviewer 1 Report
This review is on three kinds of fluorescent sensors: aptasensors, quantum dots and organic molecules. Nowadays the literature is abundant on different sensors for metal ions, including some reviews. Despite some reviews and book have been already published on the general topic of sensors, the present review might be of interest for the community since it summarizes some recent advances on sensing metal ions in aqueous media. Indeed, in literature the ions sensing is mostly proved in organic solutions in which the sensing properties are not necessarily the same than in aqueous media. Thus, the manuscript should be revised as follows:
1) The significance of the review would increase if a more detailed description of the aqueous media utilized in each cited case is reported. For instance, Figure 11 presents the sensing procedure in three different sources of water, but for most of the other cited cases there is not description on the aqueous media. Tables 1-6 should include information for each case about the volume composition of the aqueous media.
2) In the actual version of the manuscript the information on the first and last column is not correctly visualized. Such information could not reviewed by the referee
3) Table 4. It is not possible to identify the metal ion under test in each row
4) Figure 13. The sensing procedure was in ethanol. Was it a mixture with water?
5) If possible, solubility and stability of the sensors in aqueous media should be discussed. Notice that some sensors, specifically organic dyes are not soluble or stable in water.
6) Many successful cases are presented on sensing Hg2+ in aqueous media, but as it is mentioned in the review, the literature is less abundant for other metal ions. In fact, the literate is abundant in sensing and Hg2+ not only in water but also in a big variety of organic solutions and suspensions. Then, this review should briefly mention the limitations and challenges for sensing some other metals ions.
7) Figure 9. There is misprint for the units of concentration of samples a-k
8) For comparative purposes, could authors define typical values of the quantum yield of fluorescence of the presented aptasensors, quantum dots and organic molecules?
Author Response
Dear Reviewer,
thank you very much for your interesting questions and comments, which have helped us to increase the quality of this review. Please, find attached the answers to your questions.
Best regards.
Answers to Reviewer 1
This review is on three kinds of fluorescent sensors: aptasensors, quantum dots and organic molecules. Nowadays the literature is abundant on different sensors for metal ions, including some reviews. Despite some reviews and book have been already published on the general topic of sensors, the present review might be of interest for the community since it summarizes some recent advances on sensing metal ions in aqueous media. Indeed, in literature the ions sensing is mostly proved in organic solutions in which the sensing properties are not necessarily the same than in aqueous media. Thus, the manuscript should be revised as follows:
1) The significance of the review would increase if a more detailed description of the aqueous media utilized in each cited case is reported. For instance, Figure 11 presents the sensing procedure in three different sources of water, but for most of the other cited cases there is not description on the aqueous media. Tables 1-6 should include information for each case about the volume composition of the aqueous media.
Answer from the authors: Thank you very much for this interesting comment. In order to answer it, a new column has been added to every table including detailed information about the characteristics of the aqueous media in which the detection of the heavy metal ions has been carried out. That column has been highlighted in yellow in the manuscript.
Ana-lyte | ON sequence | Fluoro-phore | Quencher | Detection range | LOD | Reversi-bility | Interferent analytes | Aqueous media | Obser-vations | Ref. |
Hg2+ | 5-′FAM-CGC TTG TTT GTT CGC ACC CGT TCT TTC TT-3′ | FAM | 14.2×10-9 – 3×10-7 M | 4.28×10-9 M | Not studied | Negligible influence | Tris–HCl buffer (10 mM, pH 8.5) | [65] | ||
Hg2+ | 5′-NH2–(CH2)6– TTCTTTCTTCCCTTGTTTGTT | SYBR Green I | 1×10-9 – 1×10-2 M | Not studied | 93 – 110 % | Not studied | Tris nitrate buffer (pH 8.0, 20 mM) | [79] | ||
Hg2+ | 5′-NH2-(CH2)6-TTCTTTCTTCGCGTTGTTTGTT-3′ | Graphene oxide (GO) sheets | 1×10-9 – 50×10-9 M | 9.2×10-10 M | Not studied | Negligible influence | phosphate-buffered (PBS) saline (10 mM , pH=7.0) | [80] | ||
Hg2+ | 5′-NH2-TTCTTCCCCTTGTT-3′ | graphite carbon nitride (g-C3N4) sheets | 5×10-10 – 1×10-6 M | 1.7×10-8 M | 98.3 – 110.8 % | Cu2+, Fe3+. Ag+ | Detection range, LOD and interferent analytes calculated in Tris-HCl (pH 7.6, 25 mM) buffer containing 150 mM NaCl
Reversibi-lity studied in tap water | [81] | ||
Hg2+ | Fam-5′-GGTTGGTGTGGTTGG-3′-DABCYL) | FAM | DABCYL | 1×10-8 – 2×10-7 M | 5×10-9 M | 95 – 104 % | Pb2+ | Tris−aceta-te (pH 7.4, 10 mM) | [61] | |
Hg2+ | 5′-FAM-GGT-TGG-TGT-GGT-TGG-DABCYL-3′ | FAM | DABCYL | 1×10-8 – 2×10-7 M | 1×10-8 M | Not studied | Not studied | Tris-acetate buffer (pH 7.4, 10 mM) | [68] | |
Hg2+ | 5′-SH-3(CH2CH2O)6- TCATGTTTGTTTGTTGGCCCCCCTTCTTTCTTA-3′ linked to the AuNPs | Texas Red | Au NPs | 1×10-11 – 1×10-6 M | 5.1×10-11 M | Not studied | Negligible influence | phosphate-buffered saline (10 mM , pH=7.0) with 0.3 M NaCl | cDNA linked to the Texas Red | [64] |
Hg2+ | 5′-SH (CH2)6A10TTCTTTCTTCCCCTTGTTTGTT-FAM-3′ | FAM | Au NPs | 2×10-8 – 1×10-6 M | 1.6×10-10 M | Not studied | Negligible influence | Tris–HCl buffer (25 mM, pH 8.2) containing 0.3 M NaCl | Aptamer linked to the Au NPs at the 5’ termini | [29] |
Hg2+ | 5′ NH2- C6-CTA CAG TTT CAC CTT TTC CCC CGT TTT GGT GTT T-3′ linked to the NaYF4:Tm3+,Yb3+ UCNPs | NaYF4:Tm3+,Yb3+ UCNPs | Au NPs | 2×10-7 – 2×10-5 M | 6×10-8 M | 95.2 – 108.2 % | Negligible influence | Detection range and LOD studied in phosphate-buffered saline (10 mM, pH = 7.4) and reversibility analyzed in milk and tap water | cDNA linked to the Au NPs | [74] |
Hg2+ | 5’-NH2-TCATCGTTCTTTCTTCCCCTTGTTTGTT-3’ linked to the UCNPs | Mn2+-doped NaYF4: Yb, Er UCNPs | Au nanoballs | 5×10-8 – 5×10-7 M | 1.5×10-10 M | 91.4 – 102.3 % | Negligible influence | 25 g of real samples of shrimps or fish dipped in 225 ml of PBS (pH 7.4) | cDNA-functionalized Au nanoballs | [82] |
Hg2+ | 5′-SH–C6–TACAG TTTCA CCTTT TCCCC CGTTT TGGTG TTT-3′ linked to Au NPs | Mn:CdS/ZnS QDs | Au NPs | 1×10-9 – 1×10-6 M | 1.8×10-10 M | Not studied | Negligible influence | Tris–HCl (pH 7.4, 10 mM) buffer with 100 mM KCl and 1 mM MgCl2 | cDNA: 5′SH–C6–TGAAA CTGTA-3′ linked to Mn:CdS/ZnS | [73] |
Hg2+ | 5′-SH-CGTCTTGTCGA-3′ linked to QDs | Mn-doped CdS/ZnS core/shell QDs | Au NPs | 1×10-9 - 1×10-8 M | 4.9×10-10 M | Not studied | Negligible influence | PBS buffer (10 mM, pH 7.4) | cDNA: 5′-SH-TCGTCTTGTCG-3′ linked to the Au NPs | [83] |
Hg2+ | 5′-NH2-(CH2)6-TTCTTTCTTCGCGTTGTTTGTT-3′ labeled to the CDs | CQDs | GO | 5×10-9 – 2×10-7 M | 2.6×10-9 M | 94.7 - 109.8 % | Equal amount of Fe2+ | PBS (10 mM, pH 8.0) | [84] |
Table 1. Hg2+ fluorescent sensors based on aptamer detection
Ana-lyte | ON sequence | Fluoro-phore | Quen-cher | Detection range | LOD | Reversi-bility | Interferent analytes | Aqueous media | Obser-vations | Ref. |
Pb2+ | 5′-Cy5.5-(CH2)6-GGGTGGGTGGGTGGGT-3′ | Cy5.5 | 1×10-9 – 3×10-7 M | 2.2×10-10 M | 80 - 105 % | Negligible influence | PBS (10 mM, pH 7.4) | cDNA: 5′-NH2-(CH2)6-TTTTTTACCCACCCACCC-3′ | [75] | |
Pb2+ | 5’-GTGGGTAGGGCGGGTTGG-3’ | SYBR Green 1 | 1×10-8 – 1×10-6 M | Not studied | 98 - 102.3 % | Not studied | Tris–HAc buffer (10 mM, pH 7.4) | [85] | ||
Pb2+ | 5’-GGT TGG TGT GGT TGG-3’ | PicoGreen (PG) | 5×10-8 – 5×10-6 M | 4.8×10-9 M | Not studied | Negligible influence | Water | cDNA: 5’-CCA ACC ACA CCA ACC-3’ | [86] | |
Pb2+ | FAM-5′-GGTTGGTGTGGTTGG-3′-DABCYL) | FAM | DABCYL | 5×10-10 – 3×10-8 M | 3×10-10 M | 95 – 104 % | Hg2+ | Tris−aceta-te (pH 7.4, 10 mM) | [61] | |
Pb2+ | 5'-FAM-GGTTGGTGTGGTTGG-3' | FAM | Au NPs | 1.25×10-8 – 1×10-7 M | 1×10-8 M | 92-112 % | Slightly affected by Cu2+, Al2+ and Hg2+ | Tris–HAc buffer (5 mM, pH 7.4) | [87] | |
Pb2+ | 5′-SH-3(CH2CH2O)6-GGAAGGTGTGGAAGG-3′ linked to the Au NPs | Cy5.5 | Au NPs | 1×10-11 – 1×10-6 M | 2.7×10-13 M | Not studied | Negligible influence | phosphate-buffered saline (10 mM , pH 7) with 0.3 M NaCl | cDNA linked to Cy5.5 | [64] |
Pb2+ | 5′-ATTO647N-GGGTGGGTGGGTGGGT-3′ | ATTO647N | SWNTs | 0 – 1×10-6 M | 4.2×10-10 M | Not studied | Negligible influence | PBS buffer (10 mM, pH 7) with 0.25 M NaCl | [88] | |
Pb2+ | 5’-NH2-GGGTGGGTGGGTGGGT-3’ linked to NaYF4: Yb, Ho UCNPs | NaYF4: Yb, Ho UCNPs | Au NRs | 1×10-10 – 1×10-7 M | 5×10-11 M | 96.3 – 110.6 % | Negligible influence | 25 g of real samples of shrimps or fish dipped in 225 ml of PBS (pH 7.4) | cDNA-functionalized Au NRs | [82] |
Pb2+ | 5′-NH2-(CH2)6-GGGTGGGTGGGTGGGT-3′ | Graphene QDs | GOx | 6×10-10 – 4×10-7 M | 6×10-10 M | Not studied | Negligible influence | PBS buffer (10.0 mM, pH 7.4) | [89] | |
Pb2+ | 5′–NH2–(CH2)6–GGGTGGGTGGGTGGGT–3′ | CdSe/ZnS QDs | GO sheets | 1×10-10 – 1×10-8 M | 9×10-11 M | Not studied | Negligible influence | PBS buffer (10 mM, pH 7.4) | [90] | |
Pb2+ | 5′-GGTTGGTGTGGTTGG-3′ | perylenetetracarboxylic acid diimide (PTCDI) | 4.8×10-10 – 4.8×10-5 M | 4.8×10-10 M | 77.2–93.4 % | Negligible influence | MOPS buffer (5 mM , pH 7) | [91] | ||
Pb2+ | 5′-/3ThioMC3-D/CGATAACTCACTATrAGGAAGAGATG-3′ linked to the GQDs | Graphene QDs | Au NPs | 5×10-8 – 4×10-6 M | 1.67×10-8 M | Not studied | Negligible influence | PBS buffer (5 mM, pH 7.4) with 0.1 M NaCl | 5′-/5AmMC6/CATCTCTTCTCCGAGCCGGTCGA-AATAGTGAGT-3′ linked to the Au NPs | [92] |
Table 2. Pb2+ fluorescent sensors based on aptamer detection
Ana-lyte | QDs | Detection range | LOD | Reversibi-lity | Interferent analytes | Aqueous media | Obser-vations | Ref. |
Hg2+ | CdTe QDs | 0 – 2×10-6 M | 6.23×10-9 M | 96.9 – 99.4 % | Negligible influence | Ultrapu-re water | [141] | |
Hg2+ | capped CdTe QDs | (TGA) 1.25×10-9 – 1x10-8 M (l-cysteine) 5×10-12 – 2.5×10-11 M | (TGA) 3.5×10-10 M (l-cysteine) 2.7×10-12 M | Not studied | (TGA) not evaluated (l-cysteine) Zn2+, Cu2+ | Ultrapu-re water | QDs capped with thioglycolic acid (TGA) or l-cysteine | [109] |
Hg2+ | Cysteamine (CA)-capped CdTe QDs | 6×10-9 – 4.5×10-7 M | 4×10-9 M | 97 - 106.4 % | 10-fold Pb2+, Cu2+ and Ag+ < 7% | acetic–acetate buffer (pH 5.0) | [107] | |
Hg2+ | l-Tryptophan-capped carbon quantum dots | 1.1×10-8 – 4×10-6 M | 1.1×10-8 M | Not studied | Negligible influence | sodium phosphate buffer (10 mM, pH 6.0) | [142] | |
Hg2+ | HPEI-g-HPGs-capped CdS QDs | 1×10-8 – 1×10-4 M | 1.5×10-8 M | Not studied | Cu2+ | Tris–HCl buffer (pH 7.4, 10 mM) | [112] | |
Hg2+ | MPA coated Mn doped ZnSe/ZnS colloidal NPs | 0 – 2×10-8 M | 1×10−10 M | Not studied | Negligible influence | PBS (10 mM, pH 7.4) | [143] | |
Hg2+ | PDDA-functionalized CdTe QDs | 6×10-9 – 1×10-6 M | 5×10-9 M | 97.5 – 103 % | Negligible influence | Double distilled water | PDDA eliminates the interference from Cu2+ and Ag+ | [144] |
Hg2+ | TU-functionalized TGA-capped CdSe/CdS QDs | 5×10-9 – 1.5×10-6 M | 2.79×10-9 M | 83.8 - 95.4 % | Not studied | PBS (pH 7.73) | [116] | |
Hg2+ | CdTe@SiO2@GQDs | 1×10-8 – 2.2×10-5 M | 3.3×10-9 M | 107.3 - 108.7 % | Fe2+, Fe3+ | PBS (10 mM, pH 7.73) | [145] | |
Hg2+ | Carbon QDs blended with Rhodamine B | 1×10-7 – 4×10-5 M | 3×10-8 M | 94.5 – 957 % | glutathione (GSH) | High purity water | [146] | |
Hg2+ | N-doped carbon QDs | 1×10-7 – 1×10-4 M | 2.3×10−8 M | 97.2 – 103.8 % | GSH | Ultrapu-re water | [147] | |
Hg2+ | N-doped carbon QDs | 0.2×10-6 – 8×10-6 M | 8.7×10−8 M | 96.6 – 105.5 % | Negligible influence | PBS (50 mM, pH 7) | Doping with N improves the selectivity | [129] |
Hg2+ | N-doped carbon QDs | 0 – 2.5×10-5 M | 2.3×10-7 M | No | Negligible influence | Ultrapu-re water | [148] | |
Hg2+ | N-dopped carbon QDs | 1×10-8 – 1×10-7 M | 2.1×10−9 M | No | Not studied | PBS (10 mM, pH 7) | [149] | |
Hg2+ | N-, S-, Co- doped carbon QDs | 0 – 2×10-5 M | 1.8×10-7 M | No | Cu2+, Ni2+ | Deionized water and filtered river water | [136] | |
Hg2+ | S- and O- doped carbon nitride QDs | 1×10-8 – 1×10-6 M | 1×10−11 M | Not studied | Negligible influence | Double distilled water and tap water | [150] | |
Hg2+ | Graphene QDs | 8×10-7 – 9×10-6 M | 1×10-7 M | Not studied | Ca2+, Zn2+, Fe2+, and Co2+ < 10% | Tris–HCl buffer (pH 8, 50 mM) | [151] | |
Hg2+ | O-rich N-doped graphene QDs | 4×10-8 – 6×10-6 M | 8.6×10-9 M | 86.7 - 103.5 % | Pb2+, Cd2+, Cu2+, and Ni2+ | Tris–HCl buffer (pH 8, 10 mM) | [138] |
Table 3. Hg2+ fluorescent sensors based on QDs
Ana-lyte | QDs | Detection range | LOD | Reversi-bility | Interferent analytes | Aqueous media | Obser-vations | Ref. |
Pb2+ | xylenol orange functionalized CdSe/CdS QDs | 5×10-8 – 6×10-6 M | 2×10−8 M | 94.8 – 103.7 % | Negligible influence | PBS (pH 6.47) | [152] | |
Pb2+ | green Au NCs covalently linked to the surface of silica NPs embedded with red QDs | 2.5×10-8 – 2.5×10-7 M | 3.5×10−9 M | 95.2 – 112.4 % | Negligible influence | PBS (50 mM, pH 6) | [153] | |
Pb2+ | S-doped graphene QDs | 1×10-7 – 1.4 ×10-4 M | 3×10−8 M | Not studied | Negligible influence | PBS (3 mM, pH 7) | [139] | |
Pb2+ | Silica-coated ZnS QDs (ZnS@SiO2 QDs) | 1×10-9 – 2.6 ×10-4 M | - | No | Cd2+ | Deionized water | [154] | |
Pb2+ | Flavonoid moiety-incorporated carbon QDs | 1×10-10 – 2×10-8 M | 5.5×10−11 M | Not studied | Negligible influence | Deionized water | [155] | |
Pb2+ | CdTe QDs | 2×10-8 – 3.6×10-6 M | 8×10−8 M | Not studied | Negligible influence | Human serum | [156] | |
Cu2+ | N-acetyl-l-cysteine capped CdHgSe QDs | 1×10-9 – 4×10-7 M | 2×10-10 M | 98.3 - 101.6 % | Ag+, Co2+, Hg2+ | PBS (pH 9) | [157] | |
Cu2+ | L-cysteine capped Mn2+-doped ZnS QDs | 7.87×10-6 – 3.15×10-4 M | 3.15×10-6 M | Not studied | Hg2+ | Phosphate buffer (pH 7) | [113] | |
Cu2+ | ligand-capped CdTe QDs (CdTe-L QDs) | (5.16 ± 0.07) × 10-8 − (1.50 ± 0.03) × 10-5 M | (1.55 ± 0.05) × 10−8 M | Not studied | Negligible influence | Tris–HCl buffer (pH 6.5, 10 mM) | [158] | |
Cu2+ | inorganic CsPbBr3 perovskite QDs | 0 – 1×10-7 M | 1×10-10 M | Not studied | Negligible influence | Hexane | [159] | |
Cu2+ | Polyethylene glycol capped ZnO QDs (PEG@ZnO QDs) | 4×10-9 – 1×10-5 M | 3.33×10−9 M | 99.6 – 104.0 % | Negligible influence | Detection range, LOD and interferent analytes in ultrapure studied water, reversibi-lity in tap water | [114] | |
Cu2+ | Water-soluble silica-coated ZnS:Mn NPs (ZnS:Mn/SiO2) | 8.16×10-8 – 4.16×10-4 M | - | 94.76 – 105.82 % | Negligible influence | Seawater | [160] | |
Fe3+ | Carbon QDs | 0 – 3×10-4 M | 13.68×10-6 M | With ascorbic acid | Negligible influence | Ultra-pure water | [161] | |
Fe3+ | CdTe QDs: (1) thioglyco-lic acid capped quantum dots (Green) (2) N-Acetyl-l-cysteine capped QDs (red) | 0 – 3.5×10-6 M | 1.4×10−8 M | Not studied | Negligible influence | Deioni-zed water | [108] | |
Fe3+ | S-doped carbon QDs | 2.5×10-5 – 5×10-3 M | 9.6×10−7 M | Not studied | Negligible influence | Ultra-pure water | It works in strongly acid (pH < 2) solutions | [135] |
Fe3+ | N-, B-, S- doped carbon dots | 3×10-7 – 5.46×10-4 M | 9×10−8 M | 97.98 – 108.55 % | Negligible influence | Tris–HCl buffer (pH 7) | [162] | |
Hg2+, Pb2+ | L-cysteine-capped CdS QDs | 1×10-9 – 4×10-9 M (Hg2+) 3×10-9 – 1.5×10-8 M (Pb2+) | 1×10−9 M (Hg2+) 1×10−7 M (Pb2+) | Not studied | Negligible influence | phosphate buffer (pH 7.4) | [163] | |
Cr(III) | ligand-coated CdTe QDs (CdTe-L QDs) | (6.78 ± 0.05) ×10-9 – (3.70 ± 0.02) ×10-6 M | (20.30 ± 0.03)×10−9 M | 98.32 – 100.50 % | Negligible influence | PBS (10 mM, pH 7) | [164] | |
Cd2+ | Green emitting CdSe QDs covalently linked onto red emitting CdTe QDs | 1×10-7 – 9×10-6 M | 2.5×10−9 M | 86.5 – 102.6 % | Negligible influence | Detection range, LOD and interferent analytes studied in Tris-EDTA. Reversibility studied in lake water and tap water | [165] |
Table 4. Fluorescent sensors for different metal ions based on QDs
Ana-lyte | Organic dye | Detection range | LOD | Reversi-bility | Interferent analytes | Aqueous media | Obser-vations | Ref. |
Hg2+ | Rhodamine B | 1 – 5×10-8 M | - | Not studied | Negligible influence | Acetonitrile | Functionalized with 5-aminoisophthalic acid diethyl ester | [193] |
Hg2+ | Rhodamine B | 0 – 7×10-5 M | - | Not studied | Zn2+, Fe2+ and Cu2+ | Water | Functionalized with glucose | [194] |
Hg2+ | non-sulfur rhodamine derivative | 0 – 5×10-6 M | 2×10-7 M | Yes | Negligible influence | Acetonitrile | Functionalized with ethylene moiety | [195] |
Hg2+ | Rhodamine B (RBAI) Rhodamine 6G (RGAI) | RBAI - 5×10-6 – 2.2×10-5 M RGAI – 7.94x10-6 – 2.5×10-5 M | RBAI – 4.23×10-6 M RGAI – 6.34×10-6 M | > 90% | Negligible influence | Detection range, LOD, reversibility and interferent analytes studied in ethanol-water (4/6 v/v, 20 mM, HEPES pH 7.4). Detection also tested in living cells and mice | Functionalized with di-Aminobenzene-phenyl Isothiocyanate | [196] |
Hg2+ | Rhodamine B derivative | 0 – 1.6×10-5 M | 2.36×10-6 M | Yes | Negligible influence | Detection range, LOD, reversibility and interferent analytes studied in deionized water. Potential application analyzed in three natural water samples. | Functionalized with NS2 –containing receptor | [185] |
Hg2+ | Rhodamine derivative | 0 – 6×10-4 M | 6.79×10-6 M | Not studied | Negligible influence | DMSO–HEPES buffer (0.02 mol/L, pH 7.4; v/v = 6:4) | Functionalized with hydroxyquinoline group | [197] |
Hg2+ | Rhod-5N | 0 – 3×10-7 M | 1.5×10-9 M | Not studied | Not studied | Milli-Q water | Functionalized with BAPTA | [187] |
Hg2+ | Rhodamine C | 4×10-7 – 5×10-6 M | 7.4×10−8 M | Yes (Na2S addition) | Negligible influence | buffered HEPES (20 mM, pH = 7.0) water-ethanol (7/3, v/v) | synthesized by the reaction of rhodamine ethylenediamine and cinnamoyl chloride | [198] |
Hg2+ | Rhodamine B derivatives RW-1, RW-2 | RW-1: 5×10-7 – 3×10-6 M RW-2: 5×10-7 – 4×10-6 M | RW-1: 2.5×10−8 M RW-2: 4.2×10−8 M | Yes | Negligible influence | 4:6 CH3OH/HEPES buffer (v/v, 10 mM, pH 7.0) | Functionalized with a spirocyclic moiety | [199] |
Hg2+ | RR1 - rhodamine–rhodanine-based | 0 – 12×10-6 M | 5×10-9 M | No | Negligible influence | water–ACN (60/40 v/v) mixture | [200] | |
Hg2+, Pb2+, Cd2+ | rhodamine 6G hydrazide | Hg2+: 1×10-5 – 5×10-5 M Pb2+: 1×10-5 – 7×10-5 M Cd2+: 1×10-5 – 9×10-5 M | Hg2+: 1.6×10−8 M Pb2+: 1.2×10−8 M Cd2+: 4.7×10−8 M | Yes: Hg2+ and Cd2+ (with EDTA) No: Pb2+ | Cu2+ and Ni2+ in the case of Cd2+ detection | HEPES buffer solution (EtOH:H2O = 9/1, 10 mM HEPES buffer, pH 7.2) | Functionalized with N-methylisatin
| [189] |
Hg2+ | Fluorescein and rhodamine B | 2.5×10-7 – 2.52×10-6 M | 2.02×10-8 M | Yes | Negligible influence | Dichlorome-thane | [201] | |
Hg2+ | Coumarine derivative | 0 – 1.4×10-5 M | - | Yes (after TPEN incubation) | Negligible influence | Deionized water | Modified with azathia crown ether moiety | [202] |
Hg2+ | rhodol-coumarin | 0 – 2.5×10-5 M | 5.5×10-9 M | Not studied | Negligible influence | MeOH-H2O (v/v = 1: 1) solution | Modified with hydrazide moiety | [203] |
Hg2+ | coumarin | 0 – 4×10-6 M | 1×10-5 M | No | Co2+, Ni2+ and Cu2+ (can me masked by using EDTA) | HEPES buffer solution (20 mM HEPES, pH 7.2, EtOH:H2O = 1: 1, v/v) | thiosemicarbazide derivative reacts with Hg2+ | [204] |
Hg2+ | dibenzo-18-crown-6-ether (DB18C6) | 1.25×10-6 – 1.2×10-4 M | 1.25×10-8 M | Not studied | Cu2+, Pb2+ | Titrisol buffer (pH 7) | [205] | |
Hg2+ | 2-((2-(vinyloxy)-naphthalen-1-yl)methylene) malononitrile | 0 – 5×10-6 M | 4.31×10-8 M | Not studied | Negligible influence | PBS buffer (10 mM, pH 7.4, containing 1% CH3CN) | [206] | |
Hg2+ | Dansyl-Met-NH2 | 1×10-8 – 6×10-6 M | 5×10-9 M | Yes | Potentital interference from Fe2+, Pb2+, Cd2+, Pd2+ | HEPES buffer (10 mM, pH 7.4). Potential application also studied in synthetic marine water | [207] |
Table 5. Hg2+ sensors for different metal ions based on organic dyes
Ana-lyte | Organic dye | Detection range | LOD | Reversi-bility | Interferent analytes | Aqueous media | Obser-vations | Ref. |
Pb2+ | Rhodamine 6G derivative | 1 × 10-8 – 1 × 10-5 M | 2.7 × 10-9 M | Yes | Negligible influence | HEPES buffer (10 mM, pH 7.4). Also tested in sea shells food. | Recognition moiety attached to the R-6G derivative | [190] |
Pb2+ | rhodamine tri methoxy benzaldehyde conjugate derivative | 0 – 1 × 10-5 M | 1.5 × 10-8 M | Not studied | Negligible influence | HEPES buffer solution (pH 7.54) | [208] | |
Pb2+ | rhodamine hydroxamate derivative | 0 – 1 × 10-5 M | 2.5 × 10-7 M | Yes (adding EDTA) | Negligible influence | HEPES buffer (10 mM, pH 6.5) | Functionalized with an acyclic diethyl iminodiacetate receptor | [188] |
Pb2+ | Coumarin | 0 – 2 × 10-5 M | 1.9 × 10-9 M | Not studied | Negligible influence | phosphate‐buffer (20 mM, 1:9 DMSO/H2O (v/v), pH 8.0) | Coumarin-trizaole-based receptor: (4‐((1‐(2‐oxo‐2H‐chromen‐4‐yl)‐1H‐1,2,3‐ triazol‐5‐yl)methoxy)‐2H‐chromen‐2‐one) | [209] |
Pb2+ | Coumarin | 6 × 10-6 – 2 × 10-5 M | 3.36 × 10-11 M | Not studied | Negligible influence | HEPES buffer solution (CH3CN:H2O = 95:5, v/v, 10 mM, pH 7.2) | Functionalized with a triazole substituted 8-hydroxyquinoline (8-HQ) receptor | [192] |
Pb2+ | BODIPY fluorophore | 5 × 10-8 – 2.5 × 10-6 M | 1.34 × 10-8 M | Not studied | Negligible influence | PBS buffer (0.1 M, pH 7.2) | Functionalized with a polyamide receptor | [210] |
Pb2+ | 1,3,6-trihydroxy xanthone | 1 × 10-5 – 2 × 10-4 M | 1.8 × 10-7 M | Not studied | - | DMSO–H2O solution (2:1 ratio, v/v) | [211] | |
Pb2+ | 2-amino-4-phenyl-4H-benzo[h]chromene-3-carbonitrile | 0 – 2 × 10-3 M | 4.14 × 10-4 M | Yes | Cd2+, Fe3+, Hg2+, Cu2+ | Methanol | [212] | |
Cu2+ | rhodamine B semicarbazide | 2 × 10-8 – 3 × 10-7 M | 1.6 × 10-7 M | Not studied | Negligible influence | Methanol–water (1:1, v/v) at pH 7 | [213] | |
Cu2+ | rhodamine hydroxamate derivative | 0 – 1.2 × 10-5 M | 5.8 × 10-7 M | Yes (Na2S addition) | Negligible influence | HEPES buffer (10 mM, pH 6.5) containing 1% CH3CN (v/v)
| Functionalized with an acyclic diethyl iminodiacetate receptor | [188] |
Cu2+ | 6,7-dihydroxy-3-(3-chlorophenyl) coumarin | 0 – 2.5 × 10-6 M | 3.3 × 10-10 M | Yes (with S2-) | Negligible influence | CH3CN/H2O (90:10, v/v) | [214] | |
Cu2+ | Fluorescein | 1 × 10-6 – 6 × 10-5 M | 3 × 10-7 M | Not studied | Negligible influence | DMSO/HEPES solution(3:1, v/v, 1 mM, pH 7.2) | Functionalized with a pyrrole moiety | [215] |
Pb2+, Cu2+ | styrylcyanine dye containing pyridine | Pb2+: 3 × 10-5 – 6 × 10-4 M Cu2+: 3 × 10-6 – 9 × 10-7 M | Pb2+: 3.41 × 10-6 M Cu2+: 1.24 × 10-6 M | Not studied | Negligible influence | CH3CN–water mixture (9 : 1, v/v) | [216] | |
Zn2+ | Fluorescein-coumarin conjugate | 0 – 1 × 10-5 M | 1 × 10-7 M | Yes | Negligible influence | HEPES buffer (water/ethanol, 1:9, v/v; 10 mM HEPES; pH 7.4) | [217] | |
Cd2+ | coumarin | 0 – 1.6 × 10-5 M | - | Not studied | Hg2+ | Deionized water | Functionalized with a dipicolylamine receptor | [177] |
Ag+ | Fluorescein | L1: 0 – 1.98 × 10-6 M L2: 0 – 4.95 × 10-6 M | L1: 4 × 10-9 M L2: 3 × 10-8 M | Yes (Na2S) | Negligible influence | Ethanol | L1: functionalized with N,S- receptor L2: functionalized with N,Se- receptor | [191] |
Pd2+ | Coumarin 460 | 0 – 1 × 10-5 M | 2.5 × 10-7 M | Not studied | Negligible influence | PBS buffer containing 1% DMSO | [218] | |
Pb2+ | Rhodamine 6G derivative | 1 × 10-8 – 1 × 10-5 M | 2.7 × 10-9 M | Yes | Negligible influence | HEPES buffer (10 mM, pH 7.4). Also tested in sea shells food. | Recognition moiety attached to the R-6G derivative | [190] |
Pb2+ | rhodamine tri methoxy benzaldehyde conjugate derivative | 0 – 1 × 10-5 M | 1.5 × 10-8 M | Not studied | Negligible influence | HEPES buffer solution (pH 7.54) | [208] | |
Pb2+ | rhodamine hydroxamate derivative | 0 – 1 × 10-5 M | 2.5 × 10-7 M | Yes (adding EDTA) | Negligible influence | HEPES buffer (10 mM, pH 6.5) | Functionalized with an acyclic diethyl iminodiacetate receptor | [188] |
Pb2+ | Coumarin | 0 – 2 × 10-5 M | 1.9 × 10-9 M | Not studied | Negligible influence | phosphate‐buffer (20 mM, 1:9 DMSO/H2O (v/v), pH 8.0) | Coumarin-trizaole-based receptor: (4‐((1‐(2‐oxo‐2H‐chromen‐4‐yl)‐1H‐1,2,3‐ triazol‐5‐yl)methoxy)‐2H‐chromen‐2‐one) | [209] |
Pb2+ | Coumarin | 6 × 10-6 – 2 × 10-5 M | 3.36 × 10-11 M | Not studied | Negligible influence | HEPES buffer solution (CH3CN:H2O = 95:5, v/v, 10 mM, pH 7.2) | Functionalized with a triazole substituted 8-hydroxyquinoline (8-HQ) receptor | [192] |
Pb2+ | BODIPY fluorophore | 5 × 10-8 – 2.5 × 10-6 M | 1.34 × 10-8 M | Not studied | Negligible influence | PBS buffer (0.1 M, pH 7.2) | Functionalized with a polyamide receptor | [210] |
Pb2+ | 1,3,6-trihydroxy xanthone | 1 × 10-5 – 2 × 10-4 M | 1.8 × 10-7 M | Not studied | - | DMSO–H2O solution (2:1 ratio, v/v) | [211] | |
Pb2+ | 2-amino-4-phenyl-4H-benzo[h]chromene-3-carbonitrile | 0 – 2 × 10-3 M | 4.14 × 10-4 M | Yes | Cd2+, Fe3+, Hg2+, Cu2+ | Methanol | [212] | |
Cu2+ | rhodamine B semicarbazide | 2 × 10-8 – 3 × 10-7 M | 1.6 × 10-7 M | Not studied | Negligible influence | Methanol–water (1:1, v/v) at pH 7 | [213] | |
Cu2+ | rhodamine hydroxamate derivative | 0 – 1.2 × 10-5 M | 5.8 × 10-7 M | Yes (Na2S addition) | Negligible influence | HEPES buffer (10 mM, pH 6.5) containing 1% CH3CN (v/v)
| Functionalized with an acyclic diethyl iminodiacetate receptor | [188] |
Cu2+ | 6,7-dihydroxy-3-(3-chlorophenyl) coumarin | 0 – 2.5 × 10-6 M | 3.3 × 10-10 M | Yes (with S2-) | Negligible influence | CH3CN/H2O (90:10, v/v) | [214] | |
Cu2+ | Fluorescein | 1 × 10-6 – 6 × 10-5 M | 3 × 10-7 M | Not studied | Negligible influence | DMSO/HEPES solution(3:1, v/v, 1 mM, pH 7.2) | Functionalized with a pyrrole moiety | [215] |
Pb2+, Cu2+ | styrylcyanine dye containing pyridine | Pb2+: 3 × 10-5 – 6 × 10-4 M Cu2+: 3 × 10-6 – 9 × 10-7 M | Pb2+: 3.41 × 10-6 M Cu2+: 1.24 × 10-6 M | Not studied | Negligible influence | CH3CN–water mixture (9 : 1, v/v) | [216] | |
Zn2+ | Fluorescein-coumarin conjugate | 0 – 1 × 10-5 M | 1 × 10-7 M | Yes | Negligible influence | HEPES buffer (water/ethanol, 1:9, v/v; 10 mM HEPES; pH 7.4) | [217] | |
Cd2+ | coumarin | 0 – 1.6 × 10-5 M | - | Not studied | Hg2+ | Deionized water | Functionalized with a dipicolylamine receptor | [177] |
Ag+ | Fluorescein | L1: 0 – 1.98 × 10-6 M L2: 0 – 4.95 × 10-6 M | L1: 4 × 10-9 M L2: 3 × 10-8 M | Yes (Na2S) | Negligible influence | Ethanol | L1: functionalized with N,S- receptor L2: functionalized with N,Se- receptor | [191] |
Pd2+ | Coumarin 460 | 0 – 1 × 10-5 M | 2.5 × 10-7 M | Not studied | Negligible influence | PBS buffer containing 1% DMSO | [218] |
Table 6. Fluorescent sensors for different metal ions based on organic dyes
2) In the actual version of the manuscript the information on the first and last column is not correctly visualized. Such information could not reviewed by the referee
Answer from the authors: Thank you very much for noticing this error. The orientation of the tables has been already changed, so all the data should be properly visualized.
3) Table 4. It is not possible to identify the metal ion under test in each row
Answer from the authors: Thank you noticing this error. The orientation of the tables has been already changed, so all the data should be properly visualized in the current version of the manuscript.
4) Figure 13. The sensing procedure was in ethanol. Was it a mixture with water?
Answer from the authors: The sensing procedure of the sensors exposed in Figure 14 (Figure 13 in the previous version of the manuscript) was carried out in ethanol, not in a mixture with water. To clarify the medium of the sensing procedure, a new column that contains the feature of the aqueous media for each case has been added in Tables 1-6.
5) If possible, solubility and stability of the sensors in aqueous media should be discussed. Notice that some sensors, specifically organic dyes are not soluble or stable in water.
Answer from the authors: There is not enough information about the solubility and the stability of the sensors in the different aqueous media. However, detailed information about the characteristics of the aqueous media has been provided for each case in Tables 1-6.
6) Many successful cases are presented on sensing Hg2+ in aqueous media, but as it is mentioned in the review, the literature is less abundant for other metal ions. In fact, the literate is abundant in sensing and Hg2+ not only in water but also in a big variety of organic solutions and suspensions. Then, this review should briefly mention the limitations and challenges for sensing some other metals ions.
Answer from the authors: It is right that most of the sensors reported in the bibliography are focused on the detection of Hg2+, mainly because of its high toxicity. Although several sensors for the monitoring of other metal ions have been developed with quantum dots and organic dyes, there is a lack of sensors based on fluorescent aptamers. In this case, the high affinity of T-rich and G-rich for Hg2+ and Pb2+, respectively, is the main limitation. Thus, developing sensitive and specific ON sequences for other metal ions is one of the main challenges. To explain this fact, the following paragraph has been added in Section 2:
“Due to the high affinity of T-rich and G-rich ON sequences to these ions, and their high toxicity even at trace levels, less attention has been played to other heavy metal ions. Thus, the development of sensitive and specific ON sequences for those ions is one of the main challenges for scientists.”
7) Figure 9. There is misprint for the units of concentration of samples a-k
Answer from the authors: Thank you for noticing this misprint. In the current version, the units of concentration have been corrected and are presented in µM. Now, in Figure 9 it can be read:
“Fluorescence spectra of 10 μg/mL carbon QDs (a) and N-carbon QDs (b) upon addition of different concentrations of Hg2+ (a –k: 0, 0.6, 0.8, 1, 2, 4, 6, 8, 10, 12, and 14 μM; a–m: 0, 0.2, 0.4, 0.6, 0.8, 1, 2, 3, 4, 5, 6, 7, and 8 μM). The linear calibration ranges of each one of the sensors are shown in the insets. Reprinted with permission from [120].”
8) For comparative purposes, could authors define typical values of the quantum yield of fluorescence of the presented aptasensors, quantum dots and organic molecules?
Answer from the authors: After analyzing in detail all the references cited in Tables 1-6, the authors consider that there is not enough information in the literature for comparing the values of the quantum yields in the manuscript. For instance, the value of the quantum yield has been reported only in 2 of the 25 sensors based on aptamers (references 84 and 89), 16 of 37 devices based on quantum dots (references 129, 135, 136, 138, 142, 143, 146, 147, 148, 150, 152, 153, 157, 155, 159 and 162) and 16 of the 34 sensors based on organic dyes (references 185, 188, 190, 191, 194, 195, 201, 203, 209, 211, 212, 213, 214, 215, 216 and 217): that is, in more than the 60% of the sensors, the quantum yield of the fluorophores is not specified. The values of the quantum yield are below 40% for quantum dots, while those of organic dyes present great variability (from 1% to 97%).

Reviewer 2 Report
General comparison of all techniques might be useful, i.e., among porphyrins, MOFS, DNA FA, QD and OD for example. Rank sensitivity together with cost/efficiency for example.
How does ODN abbreviate artificial oligonucleotide (ODN)?
All of the tabular data in this publication are not properly positioned on the page and some data were cut off. For that on page 7-9, the listing under Interferent analytes (IA) were mostly n/a. Is this because of a lack of specificity with this technique? Some comment in that regard should be presented for it would appear that this method is fraught with difficulty in real world situations perhaps.
Why were the recoveries of N-CQDs greater than 100%, line 234?
Minor grammatical errors throughout the manuscript for example in line 248. These all need careful editing.
Tables p.13-15 and 19-21 also messed up and again featured a lack of IA.
It would be useful to have one real world application of sensors to find Hg2+ and Pb2+. Exactly how is this accomplished and what advantage would these techniques offer to the commercial methods?
Author Response
Dear Reviewer,
thank you very much for your interesting questions and comments, which have helped us to increase the quality of this review. Please, find attached the answers to your questions.
Best regards.
Answers to Reviewer 2
General comparison of all techniques might be useful, i.e., among porphyrins, MOFS, DNA FA, QD and OD for example. Rank sensitivity together with cost/efficiency for example.
Answer from the authors: Thank you very much for this suggestion. A new section (Section 5) has been included in the manuscript comparing some of the most representative sensors fabricated with fluorescent aptamers, quantum dots and organic dyes with other based on porphyrins and metal-organic frameworks. The following information has been added:
“5. Comparisson between fluorescent sensors for heavy metal ions based on different materials
Although this review is focused on those sensors fabricated with fluorescent aptamers, quantum dots and organic dyes, other materials can be utilized for the detection of heavy metal ions, such as porphyrins and metal-organic frameworks (MOFs). Thus, in this section, a brief comparisson between all the materials is carried out.
As it can be observed in Table 7, the sensors developed with fluorescent aptamers and quantum dots present the lowest limits of detection, oppositely to those fabricated with MOFs. Regarding to the detection ranges, the sensors based on porphyrins and MOFs are capable of detecting heavy metal ions at higher concentration ranges (from nanomolar to hundreds of micromolar concentrations) than the rest of the sensors, which monitor concentrations from the picomolar range to the micromolar one. Although reversibility and specificity are not always analyzed, the obtained results are usually positive: the sensors recover their original fluorescence intensity once the contaminants are removed from the aqueous media, and the sensors are not or slightly affected by the presence of other heavy metal ions.
Type of material | Sensitive material | Analyte | Detection range | LOD | Reversibility | Interferent Analytes | Ref. |
Fluorophore-labelled aptamer | Mn2+-doped NaYF4: Yb, Er UCNPs labelled to 5’-NH2-TCATCGTTCTTTCTTCCCCTTGTTTGTT-3’ | Hg2+ | 5×10-8 – 5×10-7 M | 1.5×10-10 M | 91.4 – 102.3 % | Negligible influence | [82] |
Texas Red labelled to 5′-SH-3(CH2CH2O)6- TCATGTTTGTTTGTTGGCCCCCCTTCTTTCTTA-3′ linked to the AuNPs | Hg2+ | 1×10-11 – 1×10-6 M | 5.1×10-11 M | Not studied | Negligible influence | [64] | |
5′- Cy5.5-SH-3(CH2CH2O)6-GGAAGGTGTGGAAGG-3′ linked to the Au NPs | Pb2+ | 1×10-11 – 1×10-6 M | 2.7×10-13 M | Not studied | Negligible influence | [64] | |
Quantum dots | S- and O- doped carbon nitride QDs | Hg2+ | 1×10-8 – 1×10-6 M | 1×10−11 M | Not studied | Negligible influence | [150] |
Flavonoid moiety-incorporated carbon QDs | Pb2+ | 1×10-10 – 2×10-8 M | 5.5×10−11 M | Not studied | Negligible influence | [157] | |
Polyethylene glycol capped ZnO QDs (PEG@ZnO QDs) | Cu2+ | 4×10-9 – 1×10-5 M | 3.33×10−9 M | 99.6 – 104 % | Negligible influence | [114] | |
Organic dyes | rhodol-coumarin | Hg2+ | 0 – 2.5×10-5 M | 5.5×10-9 M | Not studied | Negligible influence | [203] |
6,7-dihydroxy-3-(3-chlorophenyl) coumarin | Cu2+ | 0 – 2.5×10-6 M | 3.3×10-10 M | Yes (with S2-) | Negligible influence | [214] | |
Rhodamine 6G derivative | Pb2+ | 1×10-8 – 1×10-5 M | 2.7×10-9 M | Yes | Negligible influence | [190] | |
Porphyrins | 5,10,15,20-tetrakis (4-sulfonatophenyl)porphyrin(TPPS) | Hg2+ | 3.3×10-8 – 3.3×10-5 M | 3.3×10-8 M | Not studied | Negligible influence | [32] |
5,10,15,20-tetrakis (N-methyl-4-pyridyl) porphyrin (TMPyP) | Hg2+ | 5×10-9 – 1×10-7 M | 1.3×10-9 M | 96 – 105 % | Slightly affected by Pb2+ | [219] | |
5,10-bis(4-aminophenyl)-15,20-diphenyl-porphyrin (BATP) | Cd2+ | 5×10-8 – 4×10-6 M | 3.2×10-8 M | Yes | Slightly affected by Cu2+ and Hg2+ | [220] | |
Metal-organic frameworks | UiO-66-PSM | Hg2+ | 0 – 7.81×10-5 M | 5.88×10-6 M | 96.9 – 100.6 % | Negligible influence | [221] |
MIL-53(Al) | Fe3+ | 3×10-6 – 2×10-4 M | 9×10-5 M | 98 – 106 % | Negligible influence | [222] | |
UiO-66-NH2 | Cd2+ | 1×10-5 – 5×10-4 M | 3.36×10-7 M | Not studied | Not studied | [223] |
Table 7. Fluorescent sensors for heavy metal ions based on different kind of materials”
1. How does ODN abbreviate artificial oligonucleotide (ODN)?
Answer from the authors: Thank you very much for noticing this misprint. In fact, ODN corresponds to oligodeoxynucleotide, while the proper abbreviation for oligonucleotide is ON. Thus, all the abbreviations have been corrected and rewritten as ON.
2. All of the tabular data in this publication are not properly positioned on the page and some data were cut off. For that on page 7-9, the listing under Interferent analytes (IA) were mostly n/a. Is this because of a lack of specificity with this technique? Some comment in that regard should be presented for it would appear that this method is fraught with difficulty in real world situations perhaps.
Answer from the authors: Thank you very much for this comment. There was a problem with the orientation of pages in which the tables were placed. With the current orientation of the tables, all the data should be properly visualized.
Regarding to the IA, there was a misprint: where it was written n/a, now it is written “Negligible influence”, meaning that the study has been carried out, but no interference with measurable influence has been found.
Here it is possible to check the tables with the new information:
Ana-lyte | ON sequence | Fluoro-phore | Quencher | Detection range | LOD | Reversi-bility | Interferent analytes | Aqueous media | Obser-vations | Ref. |
Hg2+ | 5-′FAM-CGC TTG TTT GTT CGC ACC CGT TCT TTC TT-3′ | FAM | 14.2×10-9 – 3×10-7 M | 4.28×10-9 M | Not studied | Negligible influence | Tris–HCl buffer (10 mM, pH 8.5) | [65] | ||
Hg2+ | 5′-NH2–(CH2)6– TTCTTTCTTCCCTTGTTTGTT | SYBR Green I | 1×10-9 – 1×10-2 M | Not studied | 93 – 110 % | Not studied | Tris nitrate buffer (pH 8.0, 20 mM) | [79] | ||
Hg2+ | 5′-NH2-(CH2)6-TTCTTTCTTCGCGTTGTTTGTT-3′ | Graphene oxide (GO) sheets | 1×10-9 – 50×10-9 M | 9.2×10-10 M | Not studied | Negligible influence | phosphate-buffered (PBS) saline (10 mM , pH=7.0) | [80] | ||
Hg2+ | 5′-NH2-TTCTTCCCCTTGTT-3′ | graphite carbon nitride (g-C3N4) sheets | 5×10-10 – 1×10-6 M | 1.7×10-8 M | 98.3 – 110.8 % | Cu2+, Fe3+. Ag+ | Detection range, LOD and interferent analytes calculated in Tris-HCl (pH 7.6, 25 mM) buffer containing 150 mM NaCl
Reversibi-lity studied in tap water | [81] | ||
Hg2+ | Fam-5′-GGTTGGTGTGGTTGG-3′-DABCYL) | FAM | DABCYL | 1×10-8 – 2×10-7 M | 5×10-9 M | 95 – 104 % | Pb2+ | Tris−aceta-te (pH 7.4, 10 mM) | [61] | |
Hg2+ | 5′-FAM-GGT-TGG-TGT-GGT-TGG-DABCYL-3′ | FAM | DABCYL | 1×10-8 – 2×10-7 M | 1×10-8 M | Not studied | Not studied | Tris-acetate buffer (pH 7.4, 10 mM) | [68] | |
Hg2+ | 5′-SH-3(CH2CH2O)6- TCATGTTTGTTTGTTGGCCCCCCTTCTTTCTTA-3′ linked to the AuNPs | Texas Red | Au NPs | 1×10-11 – 1×10-6 M | 5.1×10-11 M | Not studied | Negligible influence | phosphate-buffered saline (10 mM , pH=7.0) with 0.3 M NaCl | cDNA linked to the Texas Red | [64] |
Hg2+ | 5′-SH (CH2)6A10TTCTTTCTTCCCCTTGTTTGTT-FAM-3′ | FAM | Au NPs | 2×10-8 – 1×10-6 M | 1.6×10-10 M | Not studied | Negligible influence | Tris–HCl buffer (25 mM, pH 8.2) containing 0.3 M NaCl | Aptamer linked to the Au NPs at the 5’ termini | [29] |
Hg2+ | 5′ NH2- C6-CTA CAG TTT CAC CTT TTC CCC CGT TTT GGT GTT T-3′ linked to the NaYF4:Tm3+,Yb3+ UCNPs | NaYF4:Tm3+,Yb3+ UCNPs | Au NPs | 2×10-7 – 2×10-5 M | 6×10-8 M | 95.2 – 108.2 % | Negligible influence | Detection range and LOD studied in phosphate-buffered saline (10 mM, pH = 7.4) and reversibility analyzed in milk and tap water | cDNA linked to the Au NPs | [74] |
Hg2+ | 5’-NH2-TCATCGTTCTTTCTTCCCCTTGTTTGTT-3’ linked to the UCNPs | Mn2+-doped NaYF4: Yb, Er UCNPs | Au nanoballs | 5×10-8 – 5×10-7 M | 1.5×10-10 M | 91.4 – 102.3 % | Negligible influence | 25 g of real samples of shrimps or fish dipped in 225 ml of PBS (pH 7.4) | cDNA-functionalized Au nanoballs | [82] |
Hg2+ | 5′-SH–C6–TACAG TTTCA CCTTT TCCCC CGTTT TGGTG TTT-3′ linked to Au NPs | Mn:CdS/ZnS QDs | Au NPs | 1×10-9 – 1×10-6 M | 1.8×10-10 M | Not studied | Negligible influence | Tris–HCl (pH 7.4, 10 mM) buffer with 100 mM KCl and 1 mM MgCl2 | cDNA: 5′SH–C6–TGAAA CTGTA-3′ linked to Mn:CdS/ZnS | [73] |
Hg2+ | 5′-SH-CGTCTTGTCGA-3′ linked to QDs | Mn-doped CdS/ZnS core/shell QDs | Au NPs | 1×10-9 - 1×10-8 M | 4.9×10-10 M | Not studied | Negligible influence | PBS buffer (10 mM, pH 7.4) | cDNA: 5′-SH-TCGTCTTGTCG-3′ linked to the Au NPs | [83] |
Hg2+ | 5′-NH2-(CH2)6-TTCTTTCTTCGCGTTGTTTGTT-3′ labeled to the CDs | CQDs | GO | 5×10-9 – 2×10-7 M | 2.6×10-9 M | 94.7 - 109.8 % | Equal amount of Fe2+ | PBS (10 mM, pH 8.0) | [84] |
Table 1. Hg2+ fluorescent sensors based on aptamer detection
Ana-lyte | ON sequence | Fluoro-phore | Quen-cher | Detection range | LOD | Reversi-bility | Interferent analytes | Aqueous media | Obser-vations | Ref. |
Pb2+ | 5′-Cy5.5-(CH2)6-GGGTGGGTGGGTGGGT-3′ | Cy5.5 | 1×10-9 – 3×10-7 M | 2.2×10-10 M | 80 - 105 % | Negligible influence | PBS (10 mM, pH 7.4) | cDNA: 5′-NH2-(CH2)6-TTTTTTACCCACCCACCC-3′ | [75] | |
Pb2+ | 5’-GTGGGTAGGGCGGGTTGG-3’ | SYBR Green 1 | 1×10-8 – 1×10-6 M | Not studied | 98 - 102.3 % | Not studied | Tris–HAc buffer (10 mM, pH 7.4) | [85] | ||
Pb2+ | 5’-GGT TGG TGT GGT TGG-3’ | PicoGreen (PG) | 5×10-8 – 5×10-6 M | 4.8×10-9 M | Not studied | Negligible influence | Water | cDNA: 5’-CCA ACC ACA CCA ACC-3’ | [86] | |
Pb2+ | FAM-5′-GGTTGGTGTGGTTGG-3′-DABCYL) | FAM | DABCYL | 5×10-10 – 3×10-8 M | 3×10-10 M | 95 – 104 % | Hg2+ | Tris−aceta-te (pH 7.4, 10 mM) | [61] | |
Pb2+ | 5'-FAM-GGTTGGTGTGGTTGG-3' | FAM | Au NPs | 1.25×10-8 – 1×10-7 M | 1×10-8 M | 92-112 % | Slightly affected by Cu2+, Al2+ and Hg2+ | Tris–HAc buffer (5 mM, pH 7.4) | [87] | |
Pb2+ | 5′-SH-3(CH2CH2O)6-GGAAGGTGTGGAAGG-3′ linked to the Au NPs | Cy5.5 | Au NPs | 1×10-11 – 1×10-6 M | 2.7×10-13 M | Not studied | Negligible influence | phosphate-buffered saline (10 mM , pH 7) with 0.3 M NaCl | cDNA linked to Cy5.5 | [64] |
Pb2+ | 5′-ATTO647N-GGGTGGGTGGGTGGGT-3′ | ATTO647N | SWNTs | 0 – 1×10-6 M | 4.2×10-10 M | Not studied | Negligible influence | PBS buffer (10 mM, pH 7) with 0.25 M NaCl | [88] | |
Pb2+ | 5’-NH2-GGGTGGGTGGGTGGGT-3’ linked to NaYF4: Yb, Ho UCNPs | NaYF4: Yb, Ho UCNPs | Au NRs | 1×10-10 – 1×10-7 M | 5×10-11 M | 96.3 – 110.6 % | Negligible influence | 25 g of real samples of shrimps or fish dipped in 225 ml of PBS (pH 7.4) | cDNA-functionalized Au NRs | [82] |
Pb2+ | 5′-NH2-(CH2)6-GGGTGGGTGGGTGGGT-3′ | Graphene QDs | GOx | 6×10-10 – 4×10-7 M | 6×10-10 M | Not studied | Negligible influence | PBS buffer (10.0 mM, pH 7.4) | [89] | |
Pb2+ | 5′–NH2–(CH2)6–GGGTGGGTGGGTGGGT–3′ | CdSe/ZnS QDs | GO sheets | 1×10-10 – 1×10-8 M | 9×10-11 M | Not studied | Negligible influence | PBS buffer (10 mM, pH 7.4) | [90] | |
Pb2+ | 5′-GGTTGGTGTGGTTGG-3′ | perylenetetracarboxylic acid diimide (PTCDI) | 4.8×10-10 – 4.8×10-5 M | 4.8×10-10 M | 77.2–93.4 % | Negligible influence | MOPS buffer (5 mM , pH 7) | [91] | ||
Pb2+ | 5′-/3ThioMC3-D/CGATAACTCACTATrAGGAAGAGATG-3′ linked to the GQDs | Graphene QDs | Au NPs | 5×10-8 – 4×10-6 M | 1.67×10-8 M | Not studied | Negligible influence | PBS buffer (5 mM, pH 7.4) with 0.1 M NaCl | 5′-/5AmMC6/CATCTCTTCTCCGAGCCGGTCGA-AATAGTGAGT-3′ linked to the Au NPs | [92] |
Table 2. Pb2+ fluorescent sensors based on aptamer detection
3. Why were the recoveries of N-CQDs greater than 100%, line 234?
Answer from the authors: As this manuscript is a review, those values of the recoveries have been reproduced from the original paper, where no explanation about the recoveries has been given. In fact, recoveries higher than the 100% imply that the fluorescence intensity after removing the heavy metal ions is higher than before the exposition of the sensors to the contaminants.
4. Minor grammatical errors throughout the manuscript for example in line 248. These all need careful editing.
Answer from the authors: Thank you very much. In the current version, in line 248 it can be read:
“In order to enhance the properties of CQDs, they can be doped of several compounds.”
5. Tables p.13-15 and 19-21 also messed up and again featured a lack of IA.
Answer from the authors: Thank you for noticing this error. The tables were not correctly displayed. The orientation of the tables has been already changed, so all the data should be properly visualized. Furthermore, the misprints of the column of the IA have also been corrected.
Ana-lyte | QDs | Detection range | LOD | Reversibi-lity | Interferent analytes | Aqueous media | Obser-vations | Ref. |
Hg2+ | CdTe QDs | 0 – 2×10-6 M | 6.23×10-9 M | 96.9 – 99.4 % | Negligible influence | Ultrapu-re water | [141] | |
Hg2+ | capped CdTe QDs | (TGA) 1.25×10-9 – 1x10-8 M (l-cysteine) 5×10-12 – 2.5×10-11 M | (TGA) 3.5×10-10 M (l-cysteine) 2.7×10-12 M | Not studied | (TGA) not evaluated (l-cysteine) Zn2+, Cu2+ | Ultrapu-re water | QDs capped with thioglycolic acid (TGA) or l-cysteine | [109] |
Hg2+ | Cysteamine (CA)-capped CdTe QDs | 6×10-9 – 4.5×10-7 M | 4×10-9 M | 97 - 106.4 % | 10-fold Pb2+, Cu2+ and Ag+ < 7% | acetic–acetate buffer (pH 5.0) | [107] | |
Hg2+ | l-Tryptophan-capped carbon quantum dots | 1.1×10-8 – 4×10-6 M | 1.1×10-8 M | Not studied | Negligible influence | sodium phosphate buffer (10 mM, pH 6.0) | [142] | |
Hg2+ | HPEI-g-HPGs-capped CdS QDs | 1×10-8 – 1×10-4 M | 1.5×10-8 M | Not studied | Cu2+ | Tris–HCl buffer (pH 7.4, 10 mM) | [112] | |
Hg2+ | MPA coated Mn doped ZnSe/ZnS colloidal NPs | 0 – 2×10-8 M | 1×10−10 M | Not studied | Negligible influence | PBS (10 mM, pH 7.4) | [143] | |
Hg2+ | PDDA-functionalized CdTe QDs | 6×10-9 – 1×10-6 M | 5×10-9 M | 97.5 – 103 % | Negligible influence | Double distilled water | PDDA eliminates the interference from Cu2+ and Ag+ | [144] |
Hg2+ | TU-functionalized TGA-capped CdSe/CdS QDs | 5×10-9 – 1.5×10-6 M | 2.79×10-9 M | 83.8 - 95.4 % | Not studied | PBS (pH 7.73) | [116] | |
Hg2+ | CdTe@SiO2@GQDs | 1×10-8 – 2.2×10-5 M | 3.3×10-9 M | 107.3 - 108.7 % | Fe2+, Fe3+ | PBS (10 mM, pH 7.73) | [145] | |
Hg2+ | Carbon QDs blended with Rhodamine B | 1×10-7 – 4×10-5 M | 3×10-8 M | 94.5 – 957 % | glutathione (GSH) | High purity water | [146] | |
Hg2+ | N-doped carbon QDs | 1×10-7 – 1×10-4 M | 2.3×10−8 M | 97.2 – 103.8 % | GSH | Ultrapu-re water | [147] | |
Hg2+ | N-doped carbon QDs | 0.2×10-6 – 8×10-6 M | 8.7×10−8 M | 96.6 – 105.5 % | Negligible influence | PBS (50 mM, pH 7) | Doping with N improves the selectivity | [129] |
Hg2+ | N-doped carbon QDs | 0 – 2.5×10-5 M | 2.3×10-7 M | No | Negligible influence | Ultrapu-re water | [148] | |
Hg2+ | N-dopped carbon QDs | 1×10-8 – 1×10-7 M | 2.1×10−9 M | No | Not studied | PBS (10 mM, pH 7) | [149] | |
Hg2+ | N-, S-, Co- doped carbon QDs | 0 – 2×10-5 M | 1.8×10-7 M | No | Cu2+, Ni2+ | Deionized water and filtered river water | [136] | |
Hg2+ | S- and O- doped carbon nitride QDs | 1×10-8 – 1×10-6 M | 1×10−11 M | Not studied | Negligible influence | Double distilled water and tap water | [150] | |
Hg2+ | Graphene QDs | 8×10-7 – 9×10-6 M | 1×10-7 M | Not studied | Ca2+, Zn2+, Fe2+, and Co2+ < 10% | Tris–HCl buffer (pH 8, 50 mM) | [151] | |
Hg2+ | O-rich N-doped graphene QDs | 4×10-8 – 6×10-6 M | 8.6×10-9 M | 86.7 - 103.5 % | Pb2+, Cd2+, Cu2+, and Ni2+ | Tris–HCl buffer (pH 8, 10 mM) | [138] | |
Hg2+ | N-, S-doped graphene QDs | 5×10-8 – 1.5×10-5 M | 1.4×10−8 M | (96 ± 4.7) – (116 ± 3.8) % | Negligible influence | PBS buffer (100 mM, pH 7) | [140] |
Table 3. Hg2+ fluorescent sensors based on QDs
Ana-lyte | QDs | Detection range | LOD | Reversi-bility | Interferent analytes | Aqueous media | Obser-vations | Ref. |
Pb2+ | xylenol orange functionalized CdSe/CdS QDs | 5×10-8 – 6×10-6 M | 2×10−8 M | 94.8 – 103.7 % | Negligible influence | PBS (pH 6.47) | [152] | |
Pb2+ | green Au NCs covalently linked to the surface of silica NPs embedded with red QDs | 2.5×10-8 – 2.5×10-7 M | 3.5×10−9 M | 95.2 – 112.4 % | Negligible influence | PBS (50 mM, pH 6) | [153] | |
Pb2+ | S-doped graphene QDs | 1×10-7 – 1.4 ×10-4 M | 3×10−8 M | Not studied | Negligible influence | PBS (3 mM, pH 7) | [139] | |
Pb2+ | Silica-coated ZnS QDs (ZnS@SiO2 QDs) | 1×10-9 – 2.6 ×10-4 M | - | No | Cd2+ | Deionized water | [154] | |
Pb2+ | Flavonoid moiety-incorporated carbon QDs | 1×10-10 – 2×10-8 M | 5.5×10−11 M | Not studied | Negligible influence | Deionized water | [155] | |
Pb2+ | CdTe QDs | 2×10-8 – 3.6×10-6 M | 8×10−8 M | Not studied | Negligible influence | Human serum | [156] | |
Cu2+ | N-acetyl-l-cysteine capped CdHgSe QDs | 1×10-9 – 4×10-7 M | 2×10-10 M | 98.3 - 101.6 % | Ag+, Co2+, Hg2+ | PBS (pH 9) | [157] | |
Cu2+ | L-cysteine capped Mn2+-doped ZnS QDs | 7.87×10-6 – 3.15×10-4 M | 3.15×10-6 M | Not studied | Hg2+ | Phosphate buffer (pH 7) | [113] | |
Cu2+ | ligand-capped CdTe QDs (CdTe-L QDs) | (5.16 ± 0.07) × 10-8 − (1.50 ± 0.03) × 10-5 M | (1.55 ± 0.05) × 10−8 M | Not studied | Negligible influence | Tris–HCl buffer (pH 6.5, 10 mM) | [158] | |
Cu2+ | inorganic CsPbBr3 perovskite QDs | 0 – 1×10-7 M | 1×10-10 M | Not studied | Negligible influence | Hexane | [159] | |
Cu2+ | Polyethylene glycol capped ZnO QDs (PEG@ZnO QDs) | 4×10-9 – 1×10-5 M | 3.33×10−9 M | 99.6 – 104.0 % | Negligible influence | Detection range, LOD and interferent analytes in ultrapure studied water, reversibi-lity in tap water | [114] | |
Cu2+ | Water-soluble silica-coated ZnS:Mn NPs (ZnS:Mn/SiO2) | 8.16×10-8 – 4.16×10-4 M | - | 94.76 – 105.82 % | Negligible influence | Seawater | [160] | |
Fe3+ | Carbon QDs | 0 – 3×10-4 M | 13.68×10-6 M | With ascorbic acid | Negligible influence | Ultra-pure water | [161] | |
Fe3+ | CdTe QDs: (1) thioglyco-lic acid capped quantum dots (Green) (2) N-Acetyl-l-cysteine capped QDs (red) | 0 – 3.5×10-6 M | 1.4×10−8 M | Not studied | Negligible influence | Deioni-zed water | [108] | |
Fe3+ | S-doped carbon QDs | 2.5×10-5 – 5×10-3 M | 9.6×10−7 M | Not studied | Negligible influence | Ultra-pure water | It works in strongly acid (pH < 2) solutions | [135] |
Fe3+ | N-, B-, S- doped carbon dots | 3×10-7 – 5.46×10-4 M | 9×10−8 M | 97.98 – 108.55 % | Negligible influence | Tris–HCl buffer (pH 7) | [162] | |
Hg2+, Pb2+ | L-cysteine-capped CdS QDs | 1×10-9 – 4×10-9 M (Hg2+) 3×10-9 – 1.5×10-8 M (Pb2+) | 1×10−9 M (Hg2+) 1×10−7 M (Pb2+) | Not studied | Negligible influence | phosphate buffer (pH 7.4) | [163] | |
Cr(III) | ligand-coated CdTe QDs (CdTe-L QDs) | (6.78 ± 0.05) ×10-9 – (3.70 ± 0.02) ×10-6 M | (20.30 ± 0.03)×10−9 M | 98.32 – 100.50 % | Negligible influence | PBS (10 mM, pH 7) | [164] | |
Cd2+ | Green emitting CdSe QDs covalently linked onto red emitting CdTe QDs | 1×10-7 – 9×10-6 M | 2.5×10−9 M | 86.5 – 102.6 % | Negligible influence | Detection range, LOD and interferent analytes studied in Tris-EDTA. Reversibility studied in lake water and tap water | [165] |
Table 4. Fluorescent sensors for different metal ions based on QDs
Ana-lyte | Organic dye | Detection range | LOD | Reversi-bility | Interferent analytes | Aqueous media | Obser-vations | Ref. |
Hg2+ | Rhodamine B | 1 – 5×10-8 M | - | Not studied | Negligible influence | Acetonitrile | Functionalized with 5-aminoisophthalic acid diethyl ester | [193] |
Hg2+ | Rhodamine B | 0 – 7×10-5 M | - | Not studied | Zn2+, Fe2+ and Cu2+ | Water | Functionalized with glucose | [194] |
Hg2+ | non-sulfur rhodamine derivative | 0 – 5×10-6 M | 2×10-7 M | Yes | Negligible influence | Acetonitrile | Functionalized with ethylene moiety | [195] |
Hg2+ | Rhodamine B (RBAI) Rhodamine 6G (RGAI) | RBAI - 5×10-6 – 2.2×10-5 M RGAI – 7.94x10-6 – 2.5×10-5 M | RBAI – 4.23×10-6 M RGAI – 6.34×10-6 M | > 90% | Negligible influence | Detection range, LOD, reversibility and interferent analytes studied in ethanol-water (4/6 v/v, 20 mM, HEPES pH 7.4). Detection also tested in living cells and mice | Functionalized with di-Aminobenzene-phenyl Isothiocyanate | [196] |
Hg2+ | Rhodamine B derivative | 0 – 1.6×10-5 M | 2.36×10-6 M | Yes | Negligible influence | Detection range, LOD, reversibility and interferent analytes studied in deionized water. Potential application analyzed in three natural water samples. | Functionalized with NS2 –containing receptor | [185] |
Hg2+ | Rhodamine derivative | 0 – 6×10-4 M | 6.79×10-6 M | Not studied | Negligible influence | DMSO–HEPES buffer (0.02 mol/L, pH 7.4; v/v = 6:4) | Functionalized with hydroxyquinoline group | [197] |
Hg2+ | Rhod-5N | 0 – 3×10-7 M | 1.5×10-9 M | Not studied | Not studied | Milli-Q water | Functionalized with BAPTA | [187] |
Hg2+ | Rhodamine C | 4×10-7 – 5×10-6 M | 7.4×10−8 M | Yes (Na2S addition) | Negligible influence | buffered HEPES (20 mM, pH = 7.0) water-ethanol (7/3, v/v) | synthesized by the reaction of rhodamine ethylenediamine and cinnamoyl chloride | [198] |
Hg2+ | Rhodamine B derivatives RW-1, RW-2 | RW-1: 5×10-7 – 3×10-6 M RW-2: 5×10-7 – 4×10-6 M | RW-1: 2.5×10−8 M RW-2: 4.2×10−8 M | Yes | Negligible influence | 4:6 CH3OH/HEPES buffer (v/v, 10 mM, pH 7.0) | Functionalized with a spirocyclic moiety | [199] |
Hg2+ | RR1 - rhodamine–rhodanine-based | 0 – 12×10-6 M | 5×10-9 M | No | Negligible influence | water–ACN (60/40 v/v) mixture | [200] | |
Hg2+, Pb2+, Cd2+ | rhodamine 6G hydrazide | Hg2+: 1×10-5 – 5×10-5 M Pb2+: 1×10-5 – 7×10-5 M Cd2+: 1×10-5 – 9×10-5 M | Hg2+: 1.6×10−8 M Pb2+: 1.2×10−8 M Cd2+: 4.7×10−8 M | Yes: Hg2+ and Cd2+ (with EDTA) No: Pb2+ | Cu2+ and Ni2+ in the case of Cd2+ detection | HEPES buffer solution (EtOH:H2O = 9/1, 10 mM HEPES buffer, pH 7.2) | Functionalized with N-methylisatin
| [189] |
Hg2+ | Fluorescein and rhodamine B | 2.5×10-7 – 2.52×10-6 M | 2.02×10-8 M | Yes | Negligible influence | Dichlorome-thane | [201] | |
Hg2+ | Coumarine derivative | 0 – 1.4×10-5 M | - | Yes (after TPEN incubation) | Negligible influence | Deionized water | Modified with azathia crown ether moiety | [202] |
Hg2+ | rhodol-coumarin | 0 – 2.5×10-5 M | 5.5×10-9 M | Not studied | Negligible influence | MeOH-H2O (v/v = 1: 1) solution | Modified with hydrazide moiety | [203] |
Hg2+ | coumarin | 0 – 4×10-6 M | 1×10-5 M | No | Co2+, Ni2+ and Cu2+ (can me masked by using EDTA) | HEPES buffer solution (20 mM HEPES, pH 7.2, EtOH:H2O = 1: 1, v/v) | thiosemicarbazide derivative reacts with Hg2+ | [204] |
Hg2+ | dibenzo-18-crown-6-ether (DB18C6) | 1.25×10-6 – 1.2×10-4 M | 1.25×10-8 M | Not studied | Cu2+, Pb2+ | Titrisol buffer (pH 7) | [205] | |
Hg2+ | 2-((2-(vinyloxy)-naphthalen-1-yl)methylene) malononitrile | 0 – 5×10-6 M | 4.31×10-8 M | Not studied | Negligible influence | PBS buffer (10 mM, pH 7.4, containing 1% CH3CN) | [206] | |
Hg2+ | Dansyl-Met-NH2 | 1×10-8 – 6×10-6 M | 5×10-9 M | Yes | Potentital interference from Fe2+, Pb2+, Cd2+, Pd2+ | HEPES buffer (10 mM, pH 7.4). Potential application also studied in synthetic marine water | [207] |
Table 5. Hg2+ sensors for different metal ions based on organic dyes
Analyte | Organic dye | Detection range | LOD | Reversi-bility | Interferent analytes | Aqueous media | Obser-vations | Ref. |
Pb2+ | Rhodamine 6G derivative | 1 × 10-8 – 1 × 10-5 M | 2.7 × 10-9 M | Yes | Negligible influence | HEPES buffer (10 mM, pH 7.4). Also tested in sea shells food. | Recognition moiety attached to the R-6G derivative | [190] |
Pb2+ | rhodamine tri methoxy benzaldehyde conjugate derivative | 0 – 1 × 10-5 M | 1.5 × 10-8 M | Not studied | Negligible influence | HEPES buffer solution (pH 7.54) | [208] | |
Pb2+ | rhodamine hydroxamate derivative | 0 – 1 × 10-5 M | 2.5 × 10-7 M | Yes (adding EDTA) | Negligible influence | HEPES buffer (10 mM, pH 6.5) | Functionalized with an acyclic diethyl iminodiacetate receptor | [188] |
Pb2+ | Coumarin | 0 – 2 × 10-5 M | 1.9 × 10-9 M | Not studied | Negligible influence | phosphate‐buffer (20 mM, 1:9 DMSO/H2O (v/v), pH 8.0) | Coumarin-trizaole-based receptor: (4‐((1‐(2‐oxo‐2H‐chromen‐4‐yl)‐1H‐1,2,3‐ triazol‐5‐yl)methoxy)‐2H‐chromen‐2‐one) | [209] |
Pb2+ | Coumarin | 6 × 10-6 – 2 × 10-5 M | 3.36 × 10-11 M | Not studied | Negligible influence | HEPES buffer solution (CH3CN:H2O = 95:5, v/v, 10 mM, pH 7.2) | Functionalized with a triazole substituted 8-hydroxyquinoline (8-HQ) receptor | [192] |
Pb2+ | BODIPY fluorophore | 5 × 10-8 – 2.5 × 10-6 M | 1.34 × 10-8 M | Not studied | Negligible influence | PBS buffer (0.1 M, pH 7.2) | Functionalized with a polyamide receptor | [210] |
Pb2+ | 1,3,6-trihydroxy xanthone | 1 × 10-5 – 2 × 10-4 M | 1.8 × 10-7 M | Not studied | - | DMSO–H2O solution (2:1 ratio, v/v) | [211] | |
Pb2+ | 2-amino-4-phenyl-4H-benzo[h]chromene-3-carbonitrile | 0 – 2 × 10-3 M | 4.14 × 10-4 M | Yes | Cd2+, Fe3+, Hg2+, Cu2+ | Methanol | [212] | |
Cu2+ | rhodamine B semicarbazide | 2 × 10-8 – 3 × 10-7 M | 1.6 × 10-7 M | Not studied | Negligible influence | Methanol–water (1:1, v/v) at pH 7 | [213] | |
Cu2+ | rhodamine hydroxamate derivative | 0 – 1.2 × 10-5 M | 5.8 × 10-7 M | Yes (Na2S addition) | Negligible influence | HEPES buffer (10 mM, pH 6.5) containing 1% CH3CN (v/v)
| Functionalized with an acyclic diethyl iminodiacetate receptor | [188] |
Cu2+ | 6,7-dihydroxy-3-(3-chlorophenyl) coumarin | 0 – 2.5 × 10-6 M | 3.3 × 10-10 M | Yes (with S2-) | Negligible influence | CH3CN/H2O (90:10, v/v) | [214] | |
Cu2+ | Fluorescein | 1 × 10-6 – 6 × 10-5 M | 3 × 10-7 M | Not studied | Negligible influence | DMSO/HEPES solution(3:1, v/v, 1 mM, pH 7.2) | Functionalized with a pyrrole moiety | [215] |
Pb2+, Cu2+ | styrylcyanine dye containing pyridine | Pb2+: 3 × 10-5 – 6 × 10-4 M Cu2+: 3 × 10-6 – 9 × 10-7 M | Pb2+: 3.41 × 10-6 M Cu2+: 1.24 × 10-6 M | Not studied | Negligible influence | CH3CN–water mixture (9 : 1, v/v) | [216] | |
Zn2+ | Fluorescein-coumarin conjugate | 0 – 1 × 10-5 M | 1 × 10-7 M | Yes | Negligible influence | HEPES buffer (water/ethanol, 1:9, v/v; 10 mM HEPES; pH 7.4) | [217] | |
Cd2+ | coumarin | 0 – 1.6 × 10-5 M | - | Not studied | Hg2+ | Deionized water | Functionalized with a dipicolylamine receptor | [177] |
Ag+ | Fluorescein | L1: 0 – 1.98 × 10-6 M L2: 0 – 4.95 × 10-6 M | L1: 4 × 10-9 M L2: 3 × 10-8 M | Yes (Na2S) | Negligible influence | Ethanol | L1: functionalized with N,S- receptor L2: functionalized with N,Se- receptor | [191] |
Pd2+ | Coumarin 460 | 0 – 1 × 10-5 M | 2.5 × 10-7 M | Not studied | Negligible influence | PBS buffer containing 1% DMSO | [218] | |
Pb2+ | Rhodamine 6G derivative | 1 × 10-8 – 1 × 10-5 M | 2.7 × 10-9 M | Yes | Negligible influence | HEPES buffer (10 mM, pH 7.4). Also tested in sea shells food. | Recognition moiety attached to the R-6G derivative | [190] |
Pb2+ | rhodamine tri methoxy benzaldehyde conjugate derivative | 0 – 1 × 10-5 M | 1.5 × 10-8 M | Not studied | Negligible influence | HEPES buffer solution (pH 7.54) | [208] | |
Pb2+ | rhodamine hydroxamate derivative | 0 – 1 × 10-5 M | 2.5 × 10-7 M | Yes (adding EDTA) | Negligible influence | HEPES buffer (10 mM, pH 6.5) | Functionalized with an acyclic diethyl iminodiacetate receptor | [188] |
Pb2+ | Coumarin | 0 – 2 × 10-5 M | 1.9 × 10-9 M | Not studied | Negligible influence | phosphate‐buffer (20 mM, 1:9 DMSO/H2O (v/v), pH 8.0) | Coumarin-trizaole-based receptor: (4‐((1‐(2‐oxo‐2H‐chromen‐4‐yl)‐1H‐1,2,3‐ triazol‐5‐yl)methoxy)‐2H‐chromen‐2‐one) | [209] |
Pb2+ | Coumarin | 6 × 10-6 – 2 × 10-5 M | 3.36 × 10-11 M | Not studied | Negligible influence | HEPES buffer solution (CH3CN:H2O = 95:5, v/v, 10 mM, pH 7.2) | Functionalized with a triazole substituted 8-hydroxyquinoline (8-HQ) receptor | [192] |
Pb2+ | BODIPY fluorophore | 5 × 10-8 – 2.5 × 10-6 M | 1.34 × 10-8 M | Not studied | Negligible influence | PBS buffer (0.1 M, pH 7.2) | Functionalized with a polyamide receptor | [210] |
Pb2+ | 1,3,6-trihydroxy xanthone | 1 × 10-5 – 2 × 10-4 M | 1.8 × 10-7 M | Not studied | - | DMSO–H2O solution (2:1 ratio, v/v) | [211] | |
Pb2+ | 2-amino-4-phenyl-4H-benzo[h]chromene-3-carbonitrile | 0 – 2 × 10-3 M | 4.14 × 10-4 M | Yes | Cd2+, Fe3+, Hg2+, Cu2+ | Methanol | [212] | |
Cu2+ | rhodamine B semicarbazide | 2 × 10-8 – 3 × 10-7 M | 1.6 × 10-7 M | Not studied | Negligible influence | Methanol–water (1:1, v/v) at pH 7 | [213] | |
Cu2+ | rhodamine hydroxamate derivative | 0 – 1.2 × 10-5 M | 5.8 × 10-7 M | Yes (Na2S addition) | Negligible influence | HEPES buffer (10 mM, pH 6.5) containing 1% CH3CN (v/v)
| Functionalized with an acyclic diethyl iminodiacetate receptor | [188] |
Cu2+ | 6,7-dihydroxy-3-(3-chlorophenyl) coumarin | 0 – 2.5 × 10-6 M | 3.3 × 10-10 M | Yes (with S2-) | Negligible influence | CH3CN/H2O (90:10, v/v) | [214] | |
Cu2+ | Fluorescein | 1 × 10-6 – 6 × 10-5 M | 3 × 10-7 M | Not studied | Negligible influence | DMSO/HEPES solution(3:1, v/v, 1 mM, pH 7.2) | Functionalized with a pyrrole moiety | [215] |
Pb2+, Cu2+ | styrylcyanine dye containing pyridine | Pb2+: 3 × 10-5 – 6 × 10-4 M Cu2+: 3 × 10-6 – 9 × 10-7 M | Pb2+: 3.41 × 10-6 M Cu2+: 1.24 × 10-6 M | Not studied | Negligible influence | CH3CN–water mixture (9 : 1, v/v) | [216] | |
Zn2+ | Fluorescein-coumarin conjugate | 0 – 1 × 10-5 M | 1 × 10-7 M | Yes | Negligible influence | HEPES buffer (water/ethanol, 1:9, v/v; 10 mM HEPES; pH 7.4) | [217] | |
Cd2+ | coumarin | 0 – 1.6 × 10-5 M | - | Not studied | Hg2+ | Deionized water | Functionalized with a dipicolylamine receptor | [177] |
Ag+ | Fluorescein | L1: 0 – 1.98 × 10-6 M L2: 0 – 4.95 × 10-6 M | L1: 4 × 10-9 M L2: 3 × 10-8 M | Yes (Na2S) | Negligible influence | Ethanol | L1: functionalized with N,S- receptor L2: functionalized with N,Se- receptor | [191] |
Pd2+ | Coumarin 460 | 0 – 1 × 10-5 M | 2.5 × 10-7 M | Not studied | Negligible influence | PBS buffer containing 1% DMSO | [218] |
Table 6. Fluorescent sensors for different metal ions based on organic dyes
6. It would be useful to have one real world application of sensors to find Hg2+ and Pb2+. Exactly how is this accomplished and what advantage would these techniques offer to the commercial methods?
Answer from the authors: Thank you for this interesting suggestion. In fact, a real application has been included in Section 3, which is based on paper strips coated with N-, S-codoped graphene quantum dots that are capable to detect Hg2+ in waste water with no interference from other metal ions. Some of the advantages that these sensors present are that the samples to be analyzed do not need any kind of pretreatment and the possibility of continuous monitoring.
The following paragraph and Figure have been added:
In particular, N-, S-codoped graphene QDs-based paper strips have been used in real waste water for the detection of Hg2+ ions [140]: as it can be observed in Figure 10 (a), the luminescence intensity of the QDs-coated paper strips decreased as the Hg2+ concentration increased from 10 to 200 µM. Furthermore, concentrations of 100 µM of other metal ions (Fe2+, Mn2+, Cr3+, Cd2+, Co2+ and Zn2+) did not present any interference, as it is displayed in Figure 10 (b).
Figure 10. (a) Luminescence of blank paper strips and paper strips coated with N-, S-codoped graphene QDs exposed to different Hg2+ concentrations and (b) in the presence of different metal ions in a 100 µM concentration. Reprinted with permission from [140].

Reviewer 3 Report
This manuscript reviewed the recent advances in fluorescent sensors for the monitoring of heavy metal ions in water and introduced from fluorescent aptamers, quantum dots and organic dyes three aspects. However, some revisions should be made to make the paper more convincing.
1、There should be more introduction about heavy metal ions in the Introduction , including the composition, source and toxicity to human body.
2、As for the Section 2 “Fluorescent aptasensors for the detection of heavy metal ions”, the expression is not very clear, a reasonable order should be given. In addition, many statements are repetitive.
3、How do capping agents tune the selectivity and sensitivity of semiconductor QDs? The mechanism should be supplemented.
4、There are many literatures reported about graphene QDs applying to the fluorescent sensors to detect heavy metal ions. They should also be added in Section 3.
Author Response
Dear Reviewer,
thank you very much for the questions and the comments, which have helped us to increase the quality of this review.
Best regards.
Answers to Reviewer 3
This manuscript reviewed the recent advances in fluorescent sensors for the monitoring of heavy metal ions in water and introduced from fluorescent aptamers, quantum dots and organic dyes three aspects. However, some revisions should be made to make the paper more convincing.
1、There should be more introduction about heavy metal ions in the Introduction, including the composition, source and toxicity to human body.
Answer from the authors: Thank you very much for this interesting suggestion. In Section 1 (Introduction), the following text has been included:
“Among the several water pollutants, such as plastic or waste [3], chemical fertilizers or pesticides [4] and pathogens [5], heavy metal ions are known for their high toxicity [6]. Although some of them are essential nutrients (for instance, iron, zinc or cobalt), they can be toxic at higher concentrations [7]. For its part, cadmium, lead and mercury are highly poisonous even at trace levels [8][9], showing a close association to cancer or neurodegenerative diseases [10][11]. Furthermore, heavy metal ions are non-biodegradable substances [12] and they have an accumulative effect in human body [13], where they enter, typically, through the air [14], the drinks [15] and the food chain [16], in which water plays a key role. There, metal ions can be found as a result of vehicle emissions [17], batteries [18] or industrial activities [19]. Thus, their detection at low concentrations is a matter of priority for environmental protection and disease prevention as well [20]”
2、As for the Section 2 “Fluorescent aptasensors for the detection of heavy metal ions”, the expression is not very clear, a reasonable order should be given. In addition, many statements are repetitive.
Answer from the authors: Thank you very much for this comment. In order to make this title coherent with the rest of the titles, it has been rewritten as follows: “Heavy metal ions sensors based on fluorescent aptamers”.
3、How do capping agents tune the selectivity and sensitivity of semiconductor QDs? The mechanism should be supplemented.
Answer from the authors: A brief explanation about the interaction between some capping agents and Hg2+ ions and three new references have been included in Section 3. Thus, in order to include the new information, the third paragraph has been modified as follows:
“Among the semiconductor QDs, CdTe QDs have been widely employed for the monitoring of heavy metal ions [107][108]. Furthermore, their selectivity and sensitivity can be tuned by utilizing different capping agents [109], such as thioglycolic acid (TGA) or L-cysteine: in the first case, an electron transfer process occurs between the functional groups of TGA and Hg2+ ions, which quenches the luminescent intensity of the CdTe QDs. Thus, employing TGA capped CdTe QDs, it was possible to detect Hg2+ in the nanomolar range, from 1.25×10-9 M to 1×10-8 M, with a LOD of 3.5×10-10 M Hg2+, as it can be observed in Figure 8 (a). In the case of L-cysteine capped CdTe QDs, their interaction with Hg2+ ions depends on the concentration of the metal ion: for concentrations of Hg2+ in the picomolar range, these ions interact with the carboxylate moiety of the L-cysteine on the surface of CdTe QDs by electrostatic forces [110]. As a consequence, their luminescent intensity was linearly quenched by the Hg2+ ions from 5×10-12 M to 2.5×10-11 M, as it is displayed in Figure 8 (b). Furthermore, the LOD of this sensor was 2.7×10-12 M Hg2+. At higher concentrations of Hg2+, there is an electron transfer between the Hg2+ ions and the L-cysteine capped CdTe QDs [111] which induces not only a quenching of the luminescence, but also a red shift in the luminescence peak. Other QDs that show sensitivity to Hg2+ are hyperbranched-graft-copolymers-capped CdS QDs [112], L-cysteine-capped ZnS QDs [113] or polyethylene glycol-capped ZnO QDs [114].”
4、There are many literatures reported about graphene QDs applying to the fluorescent sensors to detect heavy metal ions. They should also be added in Section 3.
Answer from the authors: Although some references about graphene quantum dots had already been included in Tables 3 and 4 (references 138, 139 and 151) of the previous version of the manuscript, some information about a real application of graphene quantum dots has been added.
In Section 3, the following paragraph and figure have been included:
In particular, N-, S-codoped graphene QDs-based paper strips have been used in real waste water for the detection of Hg2+ ions [140]: as it can be observed in Figure 10 (a), the luminescence intensity of the QDs-coated paper strips decreased as the Hg2+ concentration increased from 10 to 200 µM. Furthermore, concentrations of 100 µM of other metal ions (Fe2+, Mn2+, Cr3+, Cd2+, Co2+ and Zn2+) did not present any interference, as it is displayed in Figure 10 (b).
Figure 10. (a) Luminescence of blank paper strips and paper strips coated with N-, S-codoped graphene QDs exposed to different Hg2+ concentrations and (b) in the presence of different metal ions in a 100 µM concentration. Reprinted with permission from [140].

Round 2
Reviewer 1 Report
The revised manuscript was improved. It can be accepted for publication
Reviewer 3 Report
In this revised manuscript, the content becomes more integrated and extra necessary references have been added into the paper. The paper is suggested to be accepted for publication in this journal.